# THE PATH NOT TAKEN:
# RLVR PROVABLY LEARNS OFF THE PRINCIPALS

## ABSTRACT

Reinforcement learning (RL) reliably improves LLM reasoning while appearing to change only a small fraction of parameters. We revisit this paradox and argue that the visible sparsity is not the phenomenon itself but the trace of a ***optimization bias***, where RLVR stubbornly commits updates to preferred regions that remain invariant across datasets and RL variants, ***as if guided by an implicit compass***. We propose a Three-Gate Theory to formalize this mechanism, where the Anchor Gate I shows RL induces a one-step policy-KL leash that keeps updates proximal to the base policy; This constrained update is then steered by Gate II (Model Geometry) towards lower-curvature, spectra-preserving directions, a data-invariant feature; and finally, it is filtered by Gate III (Precision), where the bfloat16 format acts as a lens that amplifies the bias by hiding micro-updates, making the underlying pattern visible as apparent sparsity. Empirically, we validate this theory with a comprehensive suite of experiments. We show that RL preserves the model's spectral structure and avoids its principal weights, in sharp contrast to SFT, which alters spectra and mainly targets those weights. Causal interventions confirm that this bias is destroyed when the model's geometry is disrupted, proving that the geometry is the steering core of the "compass". By providing the first parameter-level account of RLVR's training dynamics: RLVR learns off-principal directions in weight space, our work not only demystifies its optimization bias but also provides a new perspective of understanding RLVR. Crucially, we show that RL operates in a distinct optimization regime from SFT, directly adapting SFT-era parameter-efficient fine-tuning (PEFT) methods can be flawed, as evidenced by our case studies on advanced sparse fine-tuning and LoRA variants, motivating the design of efficient geometry-aware, RLVR-native learning algorithms.

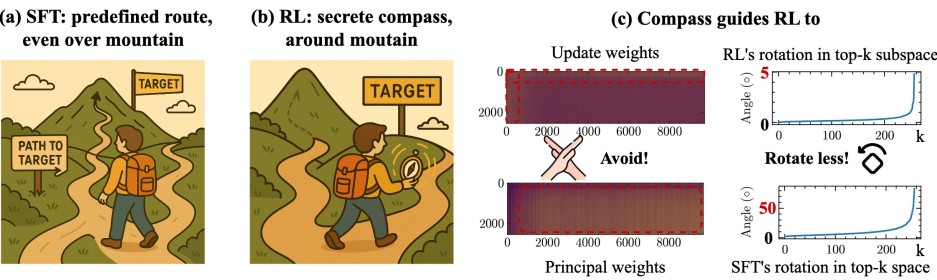

Figure 1: SFT and RL update in different manners. (a) SFT follows a predefined (guided externally) route, even over the mountain, to reach the target; (b) RLVR, without an explicit guide, behaves as if steered by an ***implicit compass (optimization bias)***, taking a detour around the mountain, with two (c) Evidences: RLVR avoids principal weights during updates (left) and rotates less in the top-$k$ subspace with preserved spectra (right).

## 1 INTRODUCTION

Large reasoning models (LRMs), such as OpenAI-o3 (Jaech et al., 2024) and DeepSeek-R1 (Guo et al., 2025), have advanced the ability of large language models to solve complex mathematical and programming tasks. A key driver is large-scale *Reinforcement Learning with Verifiable Rewards* (RLVR), which uses simple, easy-to-verify rewards to incentivize complex, multi-step reasoning. Yet, despite these advances, the mechanisms by which RL shapes model representations and behavior remain poorly understood. Given the substantial computational resources devoted to RL (relative

to SFT) and the emergence of striking new behaviors, it is natural to assume that realizing these behaviors requires substantial parameter changes. However, recent evidence points in the opposite direction: RL induces *sparse* parameter updates, whereas SFT yields *dense* ones (Mukherjee et al., 2025). This counterintuitive observation reveals a paradox: *a high-cost, high-gain process that relies on surprisingly minimal modifications.*

**Our thesis.** We resolve this paradox by uncovering a deeper mechanism behind the apparent sparsity: ***RLVR has a stubborn persistent optimization bias.*** It consistently routes visible weight updates into a narrow, reproducible subset of parameters, a pattern that remains *strikingly invariant across diverse algorithms and datasets.* This inherent selectivity is then amplified by the precision limit of bfloat16, which produces the apparent sparsity as *the symptom of a persistent optimization bias.* We refer to the organizing principle behind this optimization bias as an ***implicit RL's compass.*** As illustrated in Fig. 1, while SFT is pulled toward an explicit external target, RLVR, despite having no such teacher, is secretly guided.

These observations prompt two central questions:

*Where does this optimization bias originate, and which parameters does it preferentially update?*

In this paper, we present a **Three-Gate Theory** that formalizes this mechanism. We show that an RL update is first constrained by Gate I (Anchoring), where an on-policy KL leash keeps the policy proximal to its base. This update is then steered by Gate II (Geometry), with the intuition that the unlike a random initialized model, the pretrained model's structured landscape routes the change away from high-curvature principal subspaces under the implicit KL leash. *This geometry gate serves as a key bridge to understand why the optimization bias is data- and algorithm-invariant.* Finally, we show that the Gate III: precision of bfloat16 storage acts as a realization filter, amplifying the optimization bias by hiding minimal micro updates and leading to the apparent sparsity.

We validate this theory with a comprehensive suite of experiments, especially confirming its prediction on *which parameters it preferentially updates.* We show that RL (1) preserves the model's spectral structure, in sharp contrast to SFT; (2) avoids updating the model's principal weights (the core pathways identified by the rank-k reconstruction of the weights, defined as a key driver for parameter-efficient SFT Liu et al. (2025c); and (3) we establish causality via a geometry intervention, showing that disrupting the model's geometry with orthogonal rotations destroys the optimization bias, confirming that the geometry is the steering compass. Finally, to validate the hypothesis, we construct an "safe mask", without any training, that can closely recover the training dynamics of a dense RLVR model in terms of KL divergence wrt. base model Shenfeld et al. (2025), showing less optimization intervention, while training on the "principal weights" yields the worst trend.

Our main contributions are:

- **Observation.(Sec. 2).** For the first time, we identify a *persistent, data/algorithm-invariant optimization bias* in RLVR fine-tuning, an *implicit optimization compass* that shapes the training behaviors.

- **Theory (Sec. 3).** We propose a Three-Gate Theory (Anchor, Geometry, Precision) that provides a mechanistic account of this bias, showing how an RL update is jointly constrained, steered, and filtered.

- **Evidence (Sec. 4).** We provide strong empirical validation, consistently contrasting RL with SFT. Our evidence includes near-invariant layer spectra, sub-random overlap with principal weights, and causal interventions that confirm the role of model geometry in guiding the optimization.

- **Insight (Sec. 5).** We show that SFT-era sparse and low-rank priors (e.g., principal-targeted variants) are misaligned with RLVR's off-principal dynamics, motivating geometry-aware, RLVR-native learning algorithms.

Our results, to our knowledge, provide the first systematic link between RL training dynamics and weight-space changes, complementing concurrent analyses that remain at the abstract level (policy distribution, output KL loss) (Wu et al., 2025; Shenfeld et al., 2025). This parameter-level account also explains why RL preserves pretrained capabilities more faithfully than SFT (Wang et al., 2024; Chu et al., 2025) in large reasoning models from a fresh perspective. Furthermore, it provides a critical insight for developing efficient RL algorithms: parameter-efficient fine-tuning (PEFT) recipes must be rethought for RL, as SFT and RL show disjoint training dynamics (See Sec. 4).

Table 1: **Update sparsity in SFT vs. RLVR.** *Higher* sparsity$_{\text{bf16}}$ indicates more weights unchanged. RLVR is consistently much sparser than SFT. † *Mixed* denotes a diverse data source combining math, coding, STEM, logic puzzles, and instruction-following Liu et al. (2025a).

| Base Model | Finetuned (FT) Model | Algorithm | Data | sparsity$_{\text{bf16}}$ |
|---|---|---|---|---|
| Qwen-1.5B | DS-R1-Distill-Qwen-1.5B | SFT | Mixed | 2.8% |
| DS-R1-Distill-Qwen-1.5B | DeepScaleR-1.5B-Preview | GRPO | Math | 53.8% |
| DS-R1-Distill-Qwen-1.5B | DeepCoder-1.5B-Preview | GRPO | Code | 45.5% |
| DS-R1-Distill-Qwen-1.5B | Archer-Code-1.5B | GRPO | Code | 52.5% |
| DS-R1-Distill-Qwen-1.5B | NV-ProRL | GRPO | Mixed† | 38.4% |
| DS-R1-Distill-Qwen-1.5B | NV-ProRL-v2 | Reinforcement++ | Mixed† | 36.3% |
| Qwen3-8B-Base | Klear-Reasoner-8B-SFT | SFT | Math+Code | 0.6% |
| Klear-Reasoner-8B-SFT | Klear-Reasoner-8B | GRPO | Math+Code | 69.5% |
| Qwen3-8B-Base | GT-Qwen3-8B-Base | GRPO | Math | 79.9% |
| Qwen3-8B-Base | OURS | DAPO | Math | 79.7% |
| Qwen3-14B-Base | UniReason-Qwen3-14B-think-SFT | SFT | Math | 18.8% |
| Qwen3-14B-Base | UniReason-Qwen3-14B-RL | GRPO | Math | 68.3% |
| Qwen3-4B | Polaris-4B-Preview | DAPO | Math | 79.3% |
| DS-R1-Distill-Qwen-7B | Polaris-7B-Preview | DAPO | Math | 61.7% |
| Qwen3-30B-A3B | UloRL-A3B | GRPO | Math | 91.7% |

## 2 A STUBBORN OPTIMIZATION BIAS IN RLVR

We revisit the observation: *RL induces sparse parameter updates*, but move beyond quantification by analyzing where RL localizes these changes. Our analysis uncovers a deep "***optimization bias***" phenomenon, which we demonstrate *RL exhibits a stubborn, structured optimization bias: it consistently routes visible changes to specific regions of the network*. The observed sparsity is a readout of this bias, amplified by `bfloat16`, rather than intrinsically sparse gradients.

**Model suite.** We analyze publicly released checkpoints, as shown in Tab. 1. The suite spans multiple RLVR variants (e.g., GRPO, DAPO, Reinforcement++), diverse data domains (math, coding, instruction), and several model families and types (dense and Mixture-of-Experts). We place particular emphasis on `DeepSeek-R1-Distill-Qwen-1.5B` (`DS-Qwen-1.5B`), for which a long-horizon RL checkpoint is available (Liu et al., 2025a). This model serves as a robust case study given its extensive training for over 3,000 steps on a diverse data mixture.

### 2.1 A ROBUST, BFLOAT16-AWARE ANALYSIS OF UPDATE SPARSITY

**A `bfloat16`-aware probe for unchanged weights.** `bfloat16` (bf16) is standard in modern RL frameworks like verl (Sheng et al., 2024), to improve throughput without compromising performance. However, analyzing parameter changes under bf16 requires a careful probe. Its unique numerical format, with only 7 mantissa bits for precision, means that the smallest representable difference between two numbers scales with their magnitude. Consequently, a fixed absolute-tolerance check as used in (Mukherjee et al., 2025), is *unreliable*, which can over- or under-report (see Appendix D.1).

To ensure a rigorous report, we adopt a numerically robust, `bfloat16`-aware probe to define the update sparsity sparsity$_{\text{bf16}}$ as the fraction of parameters that remain unchanged.

**Definition 2.1** (Unchanged Weight in bf16). *Let $w_i, \widehat{w}_i \in \mathbb{R}$ be scalars stored in bf16 (finite, nonzero). We say $w_i$ is* unchanged *with respect to $\widehat{w}_i$ iff*

$$\left| \widehat{w}_i - w_i \right| \leq \eta \max\left(|w_i|, |\widehat{w}_i|\right), \qquad \eta = 10^{-3}. \tag{1}$$

*Choosing $\eta = 10^{-3} < 2^{-9}$ makes equation 1 equivalent to bitwise equality (See Appendix D.2,).*

**Definition 2.2** (bf16-aware Update Sparsity). *Write $x \approx_{\eta}^{\text{bf16}} y$ for Def. 2.1. Define the bf16 change count $\|\theta^1 - \theta^0\|_{0,\eta}^{\text{bf16}} := \left| \{ i : \theta_i^1 \not\approx_{\eta}^{\text{bf16}} \theta_i^0 \} \right|$ and the corresponding sparsity*

$$\text{sparsity}_{\text{bf16}}(\theta^0, \theta^1; \eta) := 1 - \|\theta^1 - \theta^0\|_{0,\eta}^{\text{bf16}}/n. \tag{2}$$

*where $n$ is the total number of parameters. Values near 1 indicate few stored changes, while values near 0 indicate dense apparent change.*

**RLVR update sparsity results.** As shown in Tab. 1, our analysis confirms that RL yields substantially higher update sparsity than SFT. Across models, SFT sparsity is consistently low (typically 0.6%–18.8%), whereas RL sparsity is an order of magnitude higher, ranging from 36% to 92%. However, absolute levels on recent checkpoints are lower than earlier reports (Mukherjee et al., 2025), underscoring the need for bf16-aware probes and re-evaluation on current models.

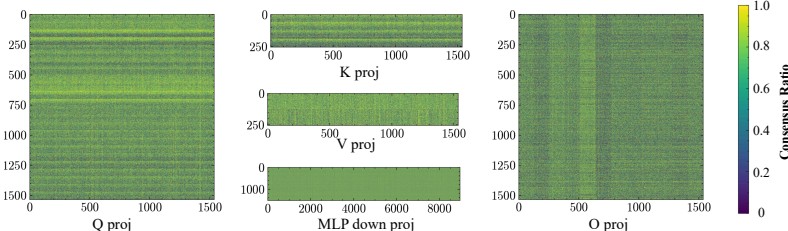

Figure 2: **Consensus ratio of weight updates** across five RLVR runs on the 13th layer's projection (Q/K/V/O) and the MLP down projection (zoom in for structures). Lighter bands indicate coordinates updated in most runs, revealing a stable, stripe-like routing pattern rather than random scatter.

## 2.2 RLVR Exhibits a Surprising Update Bias

Magnitude alone does not reveal *where* changes occur, impeding the deep analysis on *how* sparse changes arise. We therefore examine the *updated subnetwork*. We use 5 independent RLVR checkpoints from the same DS-Qwen-1.5B in Tab. 1, trained on diverse data and different RLVR algorithms. For each layer $\ell$ and run $r$, we first form the bf16-aware *changed* mask $M_\ell^{(r)} := \mathbf{1}\big[ W_\ell^{(r)} \neq_\eta^{\text{bf16}} W_\ell^0 \big]$ (Def.2.2) against the base weights $W_\ell^0$.

**Stability across runs.** We first analyze their spatial agreement using *Jaccard Overlap*. For runs $r, s$, let $A = \{(i,j) : M_{\ell,ij}^{(r)} = 1\}$ and $B = \{(i,j) : M_{\ell,ij}^{(s)} = 1\}$. We report the mean off-diagonal of the pairwise Jaccard matrix $J(A, B) = \frac{|A \cap B|}{|A \cup B|}$ and compare it to the independent Bernoulli baseline $\mathbb{E}[J] = \frac{pq}{p+q-pq}$. As summarized in Tab. 2, Jaccard is consistently high across runs, confirming a shared footprint when trained from the same base model, with Jaccard matrix shown in Fig. 8.

**Consensus ratio (*where updates land*).** Stability alone does not indicate *where* updates land. We therefore visualize and analyze the consensus ratio $C_{\ell,ij} = \frac{1}{R} \sum_{r=1}^{R} M_{\ell,ij}^{(r)}$, the fraction of runs realizing a *weight update* at coordinate $(i, j)$. Values near 1 indicate that *all* runs consistently change that weight; values near 0 indicate that none do. As shown in Fig. 2, consensus maps reveal contiguous row/column bands, stripe-like, localized routing rather than scattered noise. Especially, there are obvious *row-wise stripes in* Q/K/V *projections and column-wise stripes in* O *projections*. *This exposes a clear **optimization bias**: RLVR consistently concentrates updates in specific regions of the parameter matrices, even though the five runs use disjoint data and RL variants.*

Table 2: Cross-run stability for 13th block.

| Layer | Jaccard Overlap | Random Baseline |
|---|---|---|
| Q | 0.580 | 0.430 |
| K | 0.580 | 0.413 |
| V | 0.597 | 0.467 |
| O | 0.552 | 0.373 |
| MLP-down | 0.585 | 0.453 |
| MLP-up | 0.578 | 0.443 |
| MLP-gate | 0.575 | 0.437 |

**Temporal stability (*how the bias emerges*).** To examine *within-run* dynamics, we track the row-wise ratio $\rho_{\ell,i}(t) = \frac{1}{n_\ell} \sum_j M_{\ell,ij}(t)$ and column-wise ratio $\kappa_{\ell,j}(t) = \frac{1}{m_\ell} \sum_i M_{\ell,ij}(t)$ across checkpoints at $t$ steps. On DS-Qwen-1.5B (training setting in Appendix C.1), the *relative* profiles $\rho_{\ell,\cdot}(t)$ and $\kappa_{\ell,\cdot}(t)$ remain aligned while overall density grows as shown in Fig. 3: peaks and troughs persist. The routing bias *emerges early* and is *reinforced* over training, indicating a temporally stable phenomenon rather than a transient artifact. Moreover, the peak is consistent with the bias structure shown in Fig. 2. We also show their remaining column-wise (Q) and row-wise (O) update

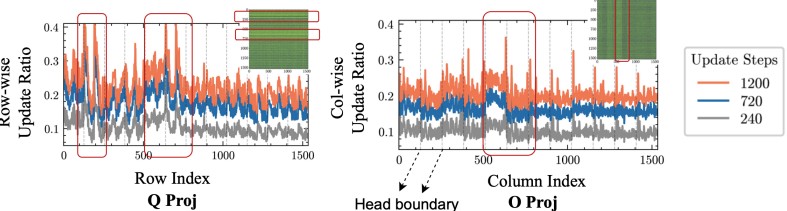

Figure 3: **Temporal emergence of the optimization bias** with row and column-wise update ratios for the 13th attention block across gradient update steps ($t \in \{240, 720, 1200\}$), smoothed with a 3-step window. The row-dominant (Q) and column-dominant (O) patterns are consistent with the bias structures in Fig. 2. We visualize the head boundaries with grey dashed lines. The bias appears not only across heads but also within heads.

ratio dynamics in Fig. 10, without a clear trend, *indicating the bias is indeed structured, not random.*

**Other model families (*whether only on Qwen*).** We observe similar stripe-structured footprints on Llama and Mistral (Fig. 9 in Appendix), suggesting the routing bias is generic to RLVR. We further examine the update consensus on the Llama model across RLVR variants in Appendix G.1.

## 2.3 SPARSITY IS A SYMPTOM, NOT THE PHENOMENON

The stable footprint of *where updates land*, persisting both throughout training and in the final model, suggests the focus should move from *sparsity* itself to the *underlying optimization bias*.

We find that sparsity is actually the *readout* of this optimization bias, whose visibility is amplified by the precision limits of `bf16` storage. Because `bf16` has a limited mantissa, changes smaller than the unit-in-the-last-place (ULP) threshold (Lemma D.2) are not representable. *Therefore, if RLVR consistently routes sub-ULP updates toward a particular subset of parameters, the stored values will not change, and the result appears as sparsity.*

We test this hypothesis by increasing the learning rate to scale otherwise sub-ULP updates above the representable threshold. As predicted, the apparent update sparsity largely disappears. This directly challenges the interpretation of (Mukherjee et al., 2025) that sparsity stems from zero gradients. Instead, our results point to sparsity as a byproduct of an optimization bias interacting with finite precision. Consistent with this view, concurrent work observes that sparsity mostly vanishes under `fp32` storage (Shenfeld et al., 2025), even though task performance does not improve.

**Remark on precision.** One natural confusion is treated the bf16 as the final cause, while it is important to note that in verl, optimizer states and gradient reductions/accumulation are maintained in `float32`[1]. So the sparsity cannot show up unless the RL process is consistently biased toward where to assign visible changes throughout the training.

> *Aha Finding! — RLVR exhibits a patterned, rather than random, optimization bias toward where the visible changes land. The apparent sparsity is a direct readout of this underlying bias, an effect amplified by bf16's precision.*

## 3 A MECHANISTIC THEORY OF THE RL OPTIMIZATION BIAS

In the post-training era, reinforcement learning (RL) has been the most compute-intensive yet powerful stage xAI (2025). Paradoxically, as shown in Sec. 2, these large gains arise not through broad updates, but through selective, patterned edits that reveal a persistent optimization bias. Understanding this bias is essential to demystify how RL achieves its improvements with two central questions:

> *Where does this optimization bias originate, and what does RL preferentially optimize?*

We answer this question with a **Three-Gate Theory**. First, on-policy RL introduces a KL constraint that **anchors** the fine-tuned policy nearby. Second, the pretrained model **geometry** steers updates towards specific regions, which is finally visualized through the *lens* of **precision**.

**Notations.** We consider a large language model with parameters $\theta$, defining a conditional distribution $\pi_\theta(y \mid x)$ over possible output token sequences $y = (y_1, \ldots, y_T) \in \mathcal{Y}$ given a prompt $x \in \mathcal{X}$ from the space $\mathcal{X}$. Each sequence $y$ is composed of tokens from a vocabulary $\mathcal{V}$ of size $N$.

**RLVR objective.** Various RLVR algorithms including PPO, GRPO, DAPO, and REINFORCE++, learn a policy $\pi_\theta$ by optimizing variants by optimizing variants of a KL-regularized objective:

$$\max_\theta \mathbb{E}_{y \sim \pi_\theta(\cdot \mid x), x \sim \mathcal{X}} \big[ R(x, y) - \beta \mathrm{KL}\big(\pi_\theta(\cdot \mid x) \,\|\, \pi_{ref}(\cdot \mid x)\big) \big]. \tag{3}$$

where $\pi_{\mathrm{ref}}$ is a fixed reference policy and $\beta \geq 0$ controls the KL regularization ($\beta = 0$ recovers the clip-only variants such as DAPO). Rewards $R(x, y)$ are *verifiable* and (after normalization) *bounded* (e.g., pass/fail or execution scores). Moreover, the surrogate typically uses the token-wise importance ratio $w_t = \frac{\pi_\theta(y_t \mid x, y_{<t})}{\pi_{\mathrm{old}}(y_t \mid x, y_{<t})}$ with clipping relative to $\pi_{\mathrm{old}}$.

### 3.1 GATE I: ANCHORING VIA AN ON-POLICY KL LEASH

We first show that online policy gradient updates yield a per-step *policy* KL bound (an **anchoring** effect), which in turn limits parameter movement during the RLVR update.

**One-step surrogate.** With equation 3, a standard sequence-level online policy-gradient surrogate is

$$\mathcal{L}_{\mathrm{PG}}(\theta) = -\mathbb{E}_{x \sim \mathcal{X}, y \sim \pi_\theta(\cdot \mid x)} \big[ A^\perp(x, y) \log \pi_\theta(y \mid x) \big], \tag{4}$$

where $A^\perp$ is a (normalized) advantage estimate, optionally *shaped* by a reference-KL log-ratio term. In practice, updates are performed over mini-batches, with a collected batch of data, not in a fully on-policy manner. But the resulting error after a small step size $\Delta\theta$ is $O(\|\Delta\theta\|^2)$ (Lemma E.1).

---

[1] verl mixed-precision settings with {`reduce_type`, `buffer_dtype`}=`float32`.

**Implicit KL leash.** The KL leash emerges as policy gradient methods can be understood as a conservative projection, keeping new policy close to its starting point while reweighting it toward higher-reward outcomes, not pulling it toward a potentially distant external distribution like SFT:

**Proposition 3.1** (One-step policy-KL leash). *Let $q(\cdot \mid x)$ be a full-support reference and let $\tilde{q}_\beta(\cdot \mid x) \propto q(\cdot \mid x) \exp(R/\beta)$ denote the soft-regularized improvement oracle. Let $\theta^+$ be the parametric fit obtained by the $M$-projection of $\tilde{q}_\beta$ onto the policy class, $\theta^+ \in \arg\min_\theta D_{\mathrm{KL}}(\tilde{q}_\beta \| \pi_\theta)$. Then, for a sufficiently small one-step update,*

$$D_{\mathrm{KL}}\big(\pi_{\theta^+} \| \pi_\theta\big) \;\leq\; (1 + o(1))\, D_{\mathrm{KL}}\big(\tilde{q}_\beta \| \pi_\theta\big), \tag{5}$$

*where the $o(1)$ term vanishes as $D_{\mathrm{KL}}(\tilde{q}_\beta \| \pi_\theta) \to 0$.*

Notably, even when the explicit KL term is removed (e.g., in DAPO with $\beta = 0$), the ratio clipping trick still imposes a KL bound $O(\varepsilon^2)$ in the small-step regime (Appendix. E.2.4), confirmed empirically with a bounded KL divergence change during a DAPO run (Fig. 11).

**Weight update constraint.** Now we show the KL leash puts a constraint on weight update $\Delta W$

**Proposition 3.2** (Policy-KL leash $\Rightarrow$ weight bound). *Assume $\log \pi_\theta$ is $C^3$ and let $F(\theta)$ denote the Fisher information. If a one-step update $\theta^+ = \theta + \Delta$ satisfies $D_{\mathrm{KL}}(\pi_{\theta^+} \| \pi_\theta) \leq K$ and, on the update subspace, $F(\theta) \succeq \mu I$ for some $\mu > 0$, then for $K$ sufficiently small*

$$\|\Delta\|_{F(\theta)} \triangleq \sqrt{\Delta^\top F(\theta)\Delta} \;\leq\; \sqrt{2K}\,(1 + o(1)), \qquad \|\Delta\|_2 \;\leq\; \sqrt{\tfrac{2K}{\mu}}\,(1 + o(1)). \tag{6}$$

*Consequently, for any weight matrix block $W \subset \theta$, $\|\Delta W\|_F \leq \sqrt{2K/\mu}\,(1 + o(1))$.*

See a detailed proof for Proposition 3.1 in Appendix E.2.1 and Proposition 3.2 in Appendix E.2.2.

> ***Take-away 1: RL update imposes an implicit KL leash (anchor effect), ensuring that the per-step drift from the current policy is small.*** This aligns with recent work arguing that even the final policy is KL-proximal Wu et al. (2025); Shenfeld et al. (2025). Our focus, however, is to understand how this leash affects the weight change dynamics.

## 3.2 Gate II: Pretrained Geometry Determines *Where* a KL-Bounded Step Goes

**From Gate I to *location*.** Gate I supplies a one-step KL leash, but it does not explain *where* the step lands. We propose Gate II: the Model Geometry Gate, where we argue that unlike a randomly initialized network, a well-pretrained model possesses a highly structured geometry, e.g., spectrum statics, high-curvature directions related to reasoning performance, acts as a "***key compass***" that determines where the update is favored to be applied.

**Layerwise norm bound from the KL leash.** Let $W_0$ be a pretrained linear block, $W_+ = W_0 + \Delta W$ the post-step block, and let $F_W \succeq \mu_W I$ be a per-layer curvature proxy. If the per-layer KL budget satisfies $\frac{1}{2}\langle \mathrm{vec}\,\Delta W, F_W\,\mathrm{vec}\,\Delta W\rangle \leq \delta_W$, then (Appendix E.10)

$$\|\Delta W\|_F \leq \sqrt{\tfrac{2\delta_W}{\mu_W}}, \qquad \|\Delta W\|_2 \leq \sqrt{\tfrac{2\delta_W}{\mu_W}}. \tag{7}$$

We then show this conservative update yields three consequences making them preserve pretrained weight spectrum instead of destroying them based on weight perturbation theory Stewart (1998).

**Limited subspace rotation.** First, as shown in Theorem 3.3, the angle between the original and updated subspaces is quadratically bounded, meaning the fundamental directions are preserved.

**Theorem 3.3** (Constrained subspace rotation with Wedin's sin–$\Theta$ theorem Wedin (1972).). *For any $k$ with $\gamma_k > 0$,*

$$\max\big(\big\|\sin\Theta(U_k(W_0), U_k(W_+))\big\|_2, \big\|\sin\Theta(V_k(W_0), V_k(W_+))\big\|_2\big) \;\leq\; \frac{\|\Delta W\|_2}{\gamma_k} \;\leq\; \frac{\sqrt{2\delta_W/\mu_W}}{\gamma_k}. \tag{8}$$

**Singular value stability.** Second, the magnitudes of the principal components themselves are preserved. The change in each singular value is bounded by the norm of the update.

**Corollary 3.4** (Singular-value stability). *For each $k$,*

$$|\sigma_k(W_+) - \sigma_k(W_0)| \;\leq\; \|\Delta W\|_2 \;\leq\; \sqrt{\frac{2\delta_W}{\mu_W}}, \qquad \sum_i \big(\sigma_i(W_+) - \sigma_i(W_0)\big)^2 \;\leq\; \|\Delta W\|_F^2 \;\leq\; \frac{2\delta_W}{\mu_W}. \tag{9}$$

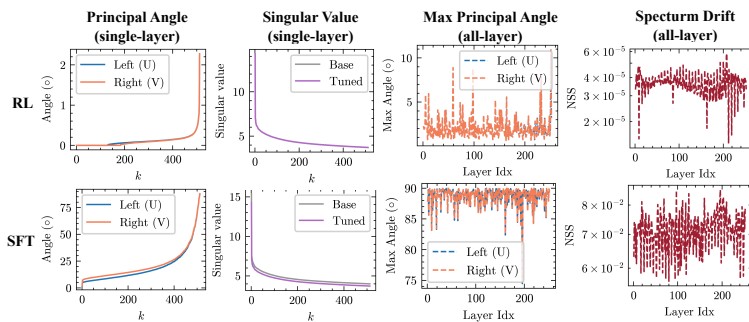

Figure 4: The spectrum probe results on the RL and SFT version on the Qwen3-8B Su et al. (2025) with the full top-$k$ principal angles and singular value curve on one exemplar layer. RLVR exhibits a surprisingly stable top-k spectrum with minimal subspace rotation and changes in top-k eigenvalues. More visualization in Appendix F.2.

**Top-k energy preservation.** Finally, these effects combine to ensure the cumulative energy of the top-k components of the weights remains stable.

**Corollary 3.5** (Top-$k$ energy and Ky Fan norms). *Let* $\|\cdot\|_{(k)} := \sum_{i=1}^{k} \sigma_i(\cdot)$ *be the Ky Fan $k$-norm. Then*

$$\big|\,\|W_+\|_{(k)} - \|W_0\|_{(k)}\,\big| \;\leq\; \sum_{i=1}^{k} \big|\sigma_i(W_+) - \sigma_i(W_0)\big| \;\leq\; k\,\|\Delta W\|_2 \;\leq\; k\sqrt{\tfrac{2\delta_W}{\mu_W}}. \tag{10}$$

See a detailed proof in Appendix E.3.

> *Take-away 2: Under the KL leash, RL updates tend to preserve the model's original weight structure rather than destroy it. This naturally favors updates in low-curvature directions of the optimization landscape, which avoids dramatic changes in model behavior.* Since directly quantifying curvature in LRM with long CoTs is computationally prohibitive, we instead adopt a powerful and efficient proxy, principal weights Liu et al. (2025c), as detailed in Sec. 4.2.

### 3.3 GATE III: PRECISION AS A LENS THAT REVEALS THE COMPASS

Building on the optimization bias, the bfloat16 with limited precision acts as a *lens*: it hides those micro-updates that occur where the RL consistently holds a weak willingness to apply large changes.

**Corollary 3.6** (Magnitude-dependent realization threshold). *A stored weight $W_{ij}$ changes at a step iff $|\Delta W_{ij}| \gtrsim \frac{1}{2}\,\mathrm{ULP}_{\mathrm{bf16}}(W_{ij})$.*

The effect of this gate has been discussed aforementioned. We would emphasize again that precision is more an *amplifier* for visible sparsity, not the *cause* of optimization bias, as optimizer states, etc., are still in float32 (See 2.3).

## 4 EMPIRICAL VALIDATION OF THE OPTIMIZATION COMPASS

We now present theory-driven empirical evidence to validate our "Optimization Compass", the core claim of Gate II: that the model's geometry is the "compass" that steers the KL-constrained updates. The following experiments confirm our theory's predictions about where the optimization bias originates and what RL preferentially optimizes.

### 4.1 RLVR PRESERVES SPECTRAL STRUCTURE, WHILE SFT DOES NOT

First, we probe spectral changes to directly test the prediction of Gate II (Geometry): the model's geometry steers the update, causing RLVR to preserve the underlying structure rather than destroy it. We examine checkpoints trained with a standard SFT-then-RLVR pipeline: one from Qwen3-8B-Base (Su et al., 2025) and another long-horizon RL run on `DS-Qwen-1.5B` (Liu et al., 2025a), both from industry with SOTA performance. Besides, we also analyze a setting where SFT and RL are applied separately to the Qwen3-14B-Base model, delivering comparable in-domain math performance (Huan et al., 2025). We compare the base weights $W_0$ with the finetuned weights $W_+$:

- **Subspace rotation.** For the top-$k$ left ($U$)/right($V$) singular subspaces, we check the rotation using **principal angles** via $\cos\theta_i(U) := \sigma_i\big(U_{0,k}^{\top}U_{+,k}\big)$ and $\cos\theta_i(V) := \sigma_i\big(V_{0,k}^{\top}V_{+,k}\big)$.

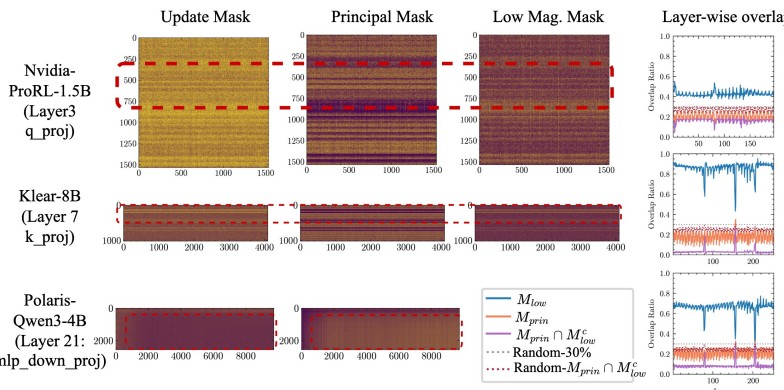

Figure 5: **RL avoids updating principal weights.** We compare the RL update mask with principal weight mask $M_{princ}$, low magnitude mask $M_{low}$, and the one $M_{princ} \cap M_{low}^c$. The layer-wise overlap between RL updates and principal weights is consistently *sub-random*, an effect more pronounced when removing its overlapped weights with $M_{low}$, i.e., $M_{princ} \cap M_{low}^c$.

- **Spectrum drift.** Beyond showing the singular value curve, we quantify singular-value change with a normalized $\ell_2$ shift: $\text{NSS}(W) = \|\sigma(W_+) - \sigma(W_0)\|_2 / \|\sigma(W_0)\|_2$

**Our findings:** *RLVR checkpoints show a surprisingly stable spectrum within their top principal components.* As shown in Fig. 4, RLVR consistently exhibits low subspace rotation and low spectrum drift. In sharp contrast, SFT induces significantly higher rotation and drift across the same metrics. This provides the first direct evidence for our "compass" theory.

## 4.2 RLVR AVOIDS PRINCIPAL WEIGHTS WHILE SFT TARGETS THEM

Next, we examine *which specific parameters the optimization compass targets or avoids* beyond a macro-level spectral check.

**Principal weights as a proxy for high-curvature directions.** Directly identifying high-curvature directions is computationally prohibitive, especially given LRM with long CoTs. Instead, we adopt a powerful proxy from recent work Liu et al. (2025c), ***principal weights,*** which is defined as *the weights with the largest magnitude after low-rank approximation*, representing its most influential computational pathways. The validity of this proxy is confirmed by their perturbation studies, which show that modifying these specific weights causes sharp *reasoning performance degradation*. This degradation is directly linked to high-curvature regions via a Taylor expansion of the loss. The *principal mask*, $M_{\text{princ}}^{(k)} = \text{Top}_\alpha\big(s_{ij}^{(k)}\big)$, is defined as the top-$\alpha$ fraction of weights with the highest score, $s_{ij}^{(k)} = |W_0^{(k)}(i,j)|$, where $W_0^k$ is the rank-$k$ SVD reconstruction of $W_0$.

**Low-magnitude weights as low-resistance pathway.** We further include the top-$\alpha$ lowest magnitude weights, as $M_{\text{low}} = \text{Bottom}_\alpha\big(|W_0|\big)$. The magnitude is also a bias from the model geometry (distribution prior), impacting how easily the weights can be updated based on our precision gate.

**Metrics.** Let $M$ be the weight update *update mask* from an RLVR run. We report the overlap ratio between our identified mask $M_\bullet$ with it, defined as $\text{Overlap}(M_\bullet, M) = \frac{|M_\bullet \cap M|}{|M|}$., with a random guess baseline overlap ratio as the density of $M_\bullet$ itself., i.e., $\alpha$.

**Our findings.** Fig. 5 visualizes the RL update mask $M$ in relation to the principal mask $M_{princ}$ and the low-magnitude mask $M_{low}$, reporting their layer-wise overlap against a random baseline as well.

The results show a clear dichotomy. RL updates exhibit a sub-random overlap with principal weights, indicating a strong tendency to avoid them. Conversely, the updates show a super-random overlap with low-magnitude weights due to their low-resistance to micro-updates. Besides, we found that the residual overlap between updates and principal weights is highly accounted for by weights that are both principal (defined by the rank-k approxima-

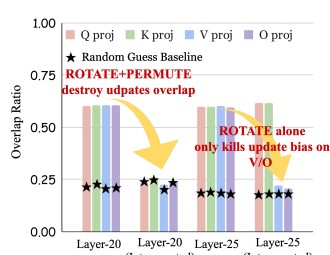

Figure 6: **Overlap ratio after intervention.**

tion of $W_0$) and low-magnitude (original $W_0$). After excluding this intersection, i.e., $M_{princ} \cap M_{low}^c$, the overlap drops significantly.

**Remark.** This leads to a crucial and counter-intuitive finding: ***RLVR and SFT are driven by updates to fundamentally different regions of a model***, though performance could be similar.

### 4.3 RLVR Relies on Model Geometry, Disrupting Geometry Destroys the Bias

Gate II posits that the pretrained model's geometry steers RL updates. To test this causal link, we deliberately "scramble" the geometry of specific layers in a Qwen3-4B-Base model using orthogonal rotations and head permutations (details in Appendix C.3) and compare the overlap $\text{Overlap}(M_\bullet, M) = \frac{|M_\bullet \cap M|}{|M|}$. between the base run with another independent run without intervention and one run with intervention.

**Findings.** We modify (i) layer 20 with ROTATE+PERMUTE, and (ii) layer 25 with ROTATE. As shown in Fig. 6, the update overlap collapsed to a random level in the intervened layers, while remaining high in all untouched layers. This provides strong causal evidence that the pretrained model's geometry is the source of the optimization bias.

### 4.4 RLVR signatures persist in agentic tasks, RLHF, robotics

**Setup.** We analyze additional *agent*, *RLHF* (RL with human feedback), and *embodied* checkpoints and apply the same weight–space diagnostics as in Sec. 4.1 and Sec. 4.2: (i) principal-subspace rotation, (ii) spectral drift, and (iii) update–principal misalignment. The extended model suite is summarized in Tab. 5. **(i)Agents.** We evaluate policies from AGENTFLOW (Li et al., 2025b) and VERL-AGENT (Feng et al., 2025) on multi-turn and long-horizon tasks. We also assess tool-augmented agents from SKYRL (Cao et al., 2025) and VERL-TOOL (Jiang et al., 2025) on *Web-Search*, *DeepSearch*, and *SWE*. **(ii)RLHF.** We include preference-optimized models trained with DPO (Rafailov et al., 2023a) and SimPO (Meng et al., 2024c), primarily targeting instruction following. **(iii)Embodied AI.** We include embodied AI models with both language and vision backbones from SimpleVLA-RL (Li et al., 2025a) and Embodied R1 (Yuan et al., 2025).

**Our Findings.** **(i) Stable spectra, minimal rotation.** Across models, top-$k$ subspaces rotate only slightly, and layer spectra remain near-identical to the base model (Fig. 15; Fig. 16; Fig. 17), matching the spectrum-preserving, off-principal regime observed earlier. **(ii) Off-principal updates.** Update masks in checkpoints consistently *avoid principal weights*: the most active bands are spatially misaligned with the principal mask (Fig. 18). **Takeaway.** RLVR's optimization dynamics, *minimal rotation, spectrum preservation, off-principal routing*, persist beyond verifiable math/code to *agents*, *embodied AI RLHF*, indicating a common, model-conditioned optimization bias within a KL-anchored RL post-training game, consistent with our Three-Gate Theory.

## 5 Theory-Guided Rethinking of Learning Algorithms for RL

A good theory should not only explain a phenomenon, beautifully validated by observations, but also provide actionable insights. Our account shows that RLVR and SFT follow disjoint optimization dynamics in parameter space, which implies that many SFT-era PEFT methods, especially those aligned with principal directions through sparse or low-rank priors, transfer poorly to RLVR. This section *validates* our predictions and *demonstrates* how they guide the redesign of learning algorithms for RL.

### 5.1 Probing Sparse Fine-Tuning in RL

Rather than judge success by final task accuracy, which is noisy and can reward "lucky" runs in RL, we instead track the token-wise **forward KL** drift $\text{KL}(\pi \parallel \pi_{\text{ref}})$ throughout training. This allows us to assess how closely a sparse run follows the dense baseline trajectory. Intuitively, if removing weights impedes the training, the policy cannot effectively shift away from the base policy.

**Masks.** We evaluate by performing RLVR on the `DS-Qwen-1.5B` with only the following weights, identified one-shot before training. (i) $U = M_{\text{princ}}$ (principal-only, sparsity 50%), (ii) $U = M_{\text{princ}}^c$ (non-principal-only, sparsity 50%), (iii) $U = M_{\text{low}}$ (lowest weights with the threshold as the mean of magnitude), (iv) $U = M_{\text{low}} \cup M_{\text{princ}}^c$ (favor non-principal and low-magnitude), and (v) a random mask with the same layer-wise sparsity as (iv). We choose 50% for (i) as we want to isolate the effect of the number of parameters for a fair comparison to see the difference between (i) and (ii).

**Results (KL in Fig. 7 and accuracy in Tab. 3)** The **union mask** $M_{\text{low}} \cup M_{\text{princ}}^c$ tracks the dense run's KL curve most closely and outperforms its random baseline, showing our theory indeed distinguishes those *highly touchable weights* with a similar trend. **principal-only** is the worst with much slower increasing training KL loss and much lower accuracy, showing a clear training block.

**Takeaways and limits.** (1) RLVR's effective updates concentrate away from *principal* directions (the ones SFT tends to favor), consistent with our theory. This reveals a fundamental contrast between SFT and RL: SFT directly targets principal weights Liu et al. (2025c), whereas RL actively avoids them. This finding demonstrates that traditional sparse fine-tuning algorithms designed for SFT may be a poor match for RL and motivates the need for a new class of RL-specific methods. (2) Freezing principal and large-magnitude weights while updating non-principal and low-magnitude ones can *approximate the dense KL trajectory* with *competitive* accuracy, demonstrating the guidance effect of our theory, especially since the mask is predicted without any additional training. (3) These are *one-shot* masks without schedule/retuning, with residual accuracy gaps expected. Future work combining our theory with dynamic mask refresh or PEFT schedules will be promising next steps (Zhao et al., 2024; Zhu et al., 2024; Liu et al., 2025c).

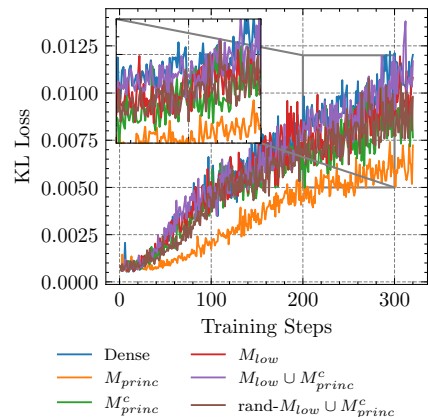

Figure 7: **KL loss curve on DS-Qwen-1.5B** under different masks. A Better visualization.

## 5.2 REVISITING LORA THROUGH THE LENS OF OUR THEORY

A recent report (Schulman & Lab, 2025) finds that low-rank LoRA, even rank-1, can match full-parameter RL performance. Our theory offers an explanation: in full-parameter RL, effective updates lie *off* the principal directions and induce only small spectral changes. Low-rank adapters can approximate these off-principal updates, while freezing the base weights regularizes training and discourages moves toward principal directions. With an appropriately scaled learning rate, the limited adapter capacity is therefore sufficient to catch up to full-parameter performance at least in the short run.

**However**, the same report suggests principal-targeted variants such as **PiSSA** (Meng et al., 2024a) should yield further gains. Our geometry account disagrees: aligning updates to top-$r$ principal directions enforces SFT-style behavior that is *misaligned* with RLVR's off-principal bias.

**Empirical test.** On DS-Qwen-1.5B with DeepMath-103K (He et al., 2025), we sweep ranks $\{8, 32, 64\}$ and learning rates $\{1 \times 10^{-4}, 5 \times 10^{-5}, 1 \times 10^{-5}\}$ for 200 steps, and report pass@1 (mean over 16 samples) on AIME24 and AMC23 (Fig. 19). To control for model effects, we repeat on Llama-3.2-3B-Instruct with a Math corpus and report pass@1 (mean over 4) on MATH500 (Fig. 20).

**Our findings.** Across settings, the principal-targeted *PiSSA* provides no clear gain over LoRA. At the higher learning rates used for low-rank adapters to match full-parameter performance, PiSSA *often becomes unstable and collapses* earlier than LoRA. This occurs because scaling the learning rate in PiSSA *enforces updates along principal directions*, higher-curvature and spectrum-distorting, precisely the directions RLVR tends to avoid. The result is brittle optimization and early collapse, whereas LoRA's off-principal updates remain better aligned with RLVR's geometry.

**Insight.** These results support the geometry-based account: principal-aligned LoRA variants are *over-fit to SFT's update geometry* and misaligned with RL's training dynamics, so success in SFT does not transfer to RL.

## 6 CONCLUSION

In this work, we resolve the paradox of sparse but effective reinforcement learning (RL) updates by identifying a persistent, geometry-aligned optimization bias, an "implicit compass" that steers training. We propose a Three-Gate Theory that provides a mechanistic account for this phenomenon, showing how on-policy constraints, pretrained model geometry, and bfloat16 precision interact to guide updates. Our experiments, including causal interventions, confirm that this compass steers RL to preserve the model's spectral structure by avoiding the principal weights targeted by Supervised Fine-Tuning (SFT). This parameter-level account not only demystifies its optimization bias but also charts a path toward a *white-box* understanding of RLVR and the design of *geometry-aware, RLVR-native* learning algorithms, rather than repurposed SFT-era heuristics.

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

## A    CLARIFICATION OF LLM USAGE

In this work, we employ LLMs to polish the writing throughout the paper and to assist in generating code for figure plotting. Besides, we use it for drawing the teaser figure.

## B    MORE RELATED WORKS

**Post-training**    Large-scale models pre-trained on broad domains serve as general-purpose backbones with extensive domain knowledge and notable zero-shot capabilities (Radford et al., 2021; Achiam et al., 2023; Touvron et al., 2023; Hu et al., 2023; Li et al., 2024; Radford et al., 2018; Brown et al., 2020). However, such pre-trained models often fail to meet the specific application requirements or align with domain-specific constraints. *Post-training* methods address this gap by adapting foundation models to downstream tasks. Common approaches include supervised fine-tuning on curated datasets (Howard & Ruder, 2018; Dodge et al., 2020; Wei et al., 2021; Chung et al., 2024), reinforcement learning from human or automated feedback (Ziegler et al., 2019; Ouyang et al., 2022; Guo et al., 2025; Zhai et al., 2024), and other recent techniques (Rafailov et al., 2023b).
Especially, the recent advances in LLM reasoning (DeepSeek-AI, 2025) highlight the effectiveness of *Reinforcement Learning with Verifiable Rewards* (RLVR), which replaces subjective human judgments with automatically verifiable signals. RLVR has been shown to significantly enhance reasoning ability using policy optimization algorithms such as PPO (Ouyang et al., 2022) and GRPO (Shao et al., 2024). Building on these successes, a growing body of work (Yu et al., 2025; Liu et al., 2025b; Luo et al., 2025a; Zhang et al., 2025; Liu et al., 2025a; Xiong et al., 2025) continues to refine RL methods tailored for LLM reasoning.

**SFT versus RL.**    Prior work comparing these paradigms has largely focused on downstream performance. A foundational result shows that on-policy RL can outperform offline SFT even with the same expert data (Ross et al., 2011). Recent empirical studies consistently reinforce this, finding that RL-tuned models often generalize better out-of-distribution (Han et al., 2025; Chu et al., 2025) and transfer more effectively to new tasks (Huan et al., 2025) than their SFT counterparts.
While these studies establish a performance hierarchy, our work investigates a different dimension: how these distinct methods affect the model's internal structure. A recent study observed that RL fine-tunes only a fraction of the network's parameters (Mukherjee et al., 2025), but this empirical finding left the underlying mechanism unexplored and did not characterize or predict the affected subnetwork. Our work aims to bridge this gap by providing a mechanistic explanation for this phenomenon.

## C    EXPERIMENTAL DETAILS

### C.1    TRAINING SETTINGS

**Models & Datasets.**    We run post-training experiments on three open models: **DeepSeek-R1-Distill-Qwen-1.5B** (Yang et al., 2024), **Qwen2.5-Math-7B** (Yang et al., 2024), and **Qwen3-Base** (Team, 2025). The maximum context length is set to 8192 for DeepSeek-R1-Distill-Qwen-1.5B and Qwen2.5-Math-7B, and to 20480for Qwen3--Base.
We evaluate primarily on mathematics using two training corpora to reduce dataset-specific confounds. (1) **DAPO+MATH (DM):** a union of the DAPO-Math-17k set[2] and the MATH dataset (Hendrycks et al., 2021). (2) **DS+SR:** the 47k DeepScaler collection (Luo et al., 2025b) combined with high-difficulty (levels 3–5) problems extracted from SimpleRL (Zeng et al., 2025).We use the version from Huan et al. (2025).

---

[2]DAPO-Math-17k

**Training details.** We implement RLVR on the VeRL pipeline (Sheng et al., 2024) and use vLLM (Kwon et al., 2023) for rollouts. We use FSDPv2 with the default mixed precision configuration. All experiments run on NVIDIA H200 GPUs. Unless otherwise noted, we use DAPO (Yu et al., 2025) *without* an explicit reference-KL penalty (ratio clipping as in DAPO), a global batch size of 256 (mini-batch 64) with 4 gradient update per step.

Per-model configurations without specific mention:

- **Qwen2.5-Math-7B** on **DM**: 16 rollouts per prompt; 8 x H200 GPUs; 300 training steps.
- **DeepSeek-R1-Distill-Qwen-1.5B** on **DS+SR**: 12 rollouts per prompt; 16 x H200 GPUs; 320 steps.
- **Qwen3-4B-Base** on **DS+SR**: 16 rollouts per prompt; 32 x H200 GPUs; 150 steps.

We optimize the actor with AdamW (Loshchilov & Hutter, 2017) (constant learning rate $1 \times 10^{-6}$, $\beta_1$=0.9, $\beta_2$=0.999). Rewards are *verifiable*: +1.0 if the extracted final answer is correct, otherwise $-1.0$ (no separate format score), following the detailed verifier implementation in Su et al. (2025). We enable an over-length penalty with an extra 1024-token budget and penalty factor 1.0.

## C.2 EVALUATION SETTINGS

We evaluate models on four widely used benchmarks: AIME24 (MAA, 2024), AIME25 (MAA, 2025), AMC23 (MAA, 2023), MATH-500 (Lightman et al., 2023), as we main train using math daastets. We used Eval-Chemy (Raoof et al., 2025) with their default temperature 0.7 and 0.8 as the top-p value. In our experiments, we used **the averaged accuracy**, i.e., $pass@1(avg@k)$ for all benchmarks. to evaluate the models' performance. Specifically, for AIME24 and AIME 25, we averaged accuracy on 64 samples, for AMC, we average accuracy on 32 samples, For MATH 500, our score is the average accuracy over 2 samples.

## C.3 INTERVENTION DETAILS

**Intervention 1: loss–preserving V/O rotation.** Let $D$ be the head dimension, $H_q$ the number of query heads, $H_{kv}$ the number of key/value heads, and $n_{\text{rep}} = H_q/H_{kv}$ (grouped GQA). Denote

$$W_v \in \mathbb{R}^{d_{\text{model}} \times (H_{kv}D)}, \qquad W_o \in \mathbb{R}^{d_{\text{model}} \times (H_qD)}.$$

Draw any orthogonal $R \in \mathbb{R}^{D \times D}$ (Haar/Hadamard) and form the block rotations

$$R_{kv} = \text{diag}(\underbrace{R, \ldots, R}_{H_{kv}}) \in \mathbb{R}^{(H_{kv}D) \times (H_{kv}D)}, \qquad R_q = \text{diag}(\underbrace{R, \ldots, R}_{n_{\text{rep}}}, \underbrace{R, \ldots, R}_{n_{\text{rep}}}, \ldots) \in \mathbb{R}^{(H_qD) \times (H_qD)}.$$

We edit the weights by right–multiplication along the head axis:

$$\boxed{W_v' = W_v R_{kv}, \qquad W_o' = W_o R_q.} \tag{11}$$

If $b_v$ exists, reshape $b_v$ per head and set $b_v' = b_v R_{kv}$.

**Proposition C.1** (Exact invariance). *Let* $\text{Ctx} = \text{Attn}(Q, K, V) \in \mathbb{R}^{\cdot \times (H_qD)}$. *Under equation 11,*

$$\text{out}' = \text{Attn}(Q, K, V R_{kv}) (W_o R_q)^\top = \text{Ctx} R_q R_q^\top W_o^\top = \text{Ctx} W_o^\top = \text{out}.$$

**Intervention 2: head shuffle (lossless).** Let $P_{kv}$ be a permutation of the $H_{kv}$ KV heads and $P_q$ its grouped expansion to $H_q$ heads. Apply

$$\text{rows of } (W_k, W_v) \leftarrow P_{kv}, \qquad \text{rows of } W_q \leftarrow P_q, \qquad \text{columns of } W_o \leftarrow P_q^{-1}.$$

$$\text{cols of } (W_k, W_v) \leftarrow P_{kv}, \qquad \text{cols of } W_q \leftarrow P_q, \qquad \text{columns of } W_o \leftarrow P_q^{-1}.$$

This relabels which head carries which subspace, while leaving the block function unchanged. We show that after weight intervention, the model weights update position has a sub-random overlap while those untouched weights stay a high overlap.

## C.4 EVALUATION SETTINGS

# D EXAMPLES OF WHY PREVIOUS IDENTIFIED METHOD FAILS

## D.1 FAILURES OF A FIXED ABSOLUTE TOLERANCE RULE

- **False positives at large scale.** Within $[2^{10}, 2^{11})$=[1024, 2048), the bf16 spacing is $\text{ULP}_{\text{bf16}} = 2^{10-7} = 8$. Numbers like 1024.001 and 1024.002 differ by $10^{-3} > 10^{-5}$, hence would be flagged as "changed" by the $10^{-5}$ rule, yet both round to the same bf16 code (1024), i.e., *no storage-level change*.

- **False negatives at small scale.** Around $10^{-6} \approx 2^{-20}$, the bf16 spacing is $\text{ULP}_{\text{bf16}} = 2^{-27} \approx 7.45 \times 10^{-9}$. Weights $w = 10^{-6}$ and $\widehat{w} = 2 \times 10^{-6}$ differ by $10^{-6} \leq 10^{-5}$ and would be marked "equal" by the $10^{-5}$ rule, yet they are separated by $\approx 134$ ULPs and quantize to *different* bf16 codes.

## D.2 Justification of our probe

**Lemma D.1** (Gap between distinct bf16 representables). *If $x \neq y$ are normalized bf16 numbers in the same binade $[2^e, 2^{e+1})$, then*

$$|x - y| \geq 2^{e-7} \quad and \quad \frac{|x - y|}{\max(|x|, |y|)} > 2^{-8}.$$

*The strict inequality also holds across the binade boundary.*

**Lemma D.2** (ULP lens: magnitude-dependent threshold). *For normalized bf16 values $x$ with $|x| \in [2^e, 2^{e+1})$,*

$$\frac{\text{ULP}_{\text{bf16}}(x)}{|x|} \in (2^{-8}, 2^{-7}] = (0.390625\%, 0.78125\%].$$

*Hence the* minimal realized *relative update at magnitude $|x|$ is $\gtrsim \frac{1}{2} \text{ULP}_{\text{bf16}}(x)/|x| \in (0.195\%, 0.391\%]$. In particular, larger $|x|$ requires a larger absolute step to register.* $\qquad\square$

**Proposition D.3** (Soundness and completeness of the probe). *Let $w_i, \widehat{w}_i$ be normalized bf16 values (finite, nonzero), and suppose $\eta < \frac{1}{2} \min_x \text{ULP}_{\text{bf16}}(x)/|x| = 2^{-9} \approx 1.953 \cdot 10^{-3}$. Then*

$$\left| \widehat{w}_i - w_i \right| \leq \eta \max(|w_i|, |\widehat{w}_i|) \quad \Longleftrightarrow \quad \text{bf16}(w_i) = \text{bf16}(\widehat{w}_i).$$

*Proof.* ($\Rightarrow$)If $w_i \neq \widehat{w}_i$, Lemma D.2 gives $|\widehat{w}_i - w_i|/\max(|w_i|, |\widehat{w}_i|) > 2^{-8} > 2\eta$, contradiction. Hence $w_i = \widehat{w}_i$ as bf16 numbers.
($\Leftarrow$) If the stored bf16 values are equal, the difference is 0, which satisfies equation 1. $\qquad\square \qquad\square$

**Corollary D.4** (Choice $\eta = 10^{-3}$ is safe). *Since $10^{-3} < 2^{-9}$, Proposition D.3 applies: the test equation 1 passes iff the two bf16 entries are bit-wise identical (or both zero). Thus $\eta = 10^{-3}$ yields a* scale-aware *probe that flags equality only when storage is unchanged.*

# E Math Analysis

## E.1 Policy-Gradient Fine-Tuning (DAPO)

Assume an *old* policy $\pi_{\text{old}}$ that we use to sample $G$ candidate completions $y^{1:G}$ for each prompt $x \in \mathcal{X}$. For a single token $y_{i,t}$ (token $t$ in completion $i$) we define the *importance-weighted advantage*

$$w_{i,t} = \underbrace{\frac{\pi_\theta(y_{i,t}|x, y_{<t})}{\pi_{\text{old}}(y_{i,t}|x, y_{<t})}}_{\text{importance ratio}} \hat{A}_{i,t} \, \mathbb{I}_{\text{clip}} \quad \in \mathbb{R}, \tag{1}$$

where $\hat{A}_{i,t}$ is the estimated advantage and $\mathbb{I}_{\text{clip}} \in \{0, 1\}$ implements the usual trust-region clipping.

**Token-level objective.** The DAPO loss can be written as a sum of weighted log-probabilities

$$J_{\text{RL}}(\theta) = \mathbb{E}_{x \sim \mathcal{X}, \, y^{1:G} \sim \pi_{\text{old}}} \left[ \frac{1}{\sum_i |y^i|} \sum_{i=1}^{G} \sum_{t=1}^{|y^i|} w_{i,t} \log \pi_\theta(y_{i,t} \mid x, y^i_{<t}) \right]. \tag{2}$$

## E.2 Proof of Gate I: On-Policy RL Implies a One-Step KL Leash

This appendix provides the standard tilting oracle and $M$-projection facts, local second-order expansions, and the proof of the one-step policy-KL leash (Prop. 3.1 in the main text). ***We keep the proof concise, otherwise too lengthy, especially for those has shown in some prior work Shenfeld et al. (2025); Wu et al. (2025).*** Our one-step analysis is inspired by recent work Wu et al. (2025); Shenfeld et al. (2025), which uses a similar variational approach to show that even the final converged policy remains KL-proximal to the base policy. We also record a trust-region/clipping bound used when $\beta = 0$.

Throughout, $x$ is fixed, $q(\cdot \mid x)$ has full support on $\mathcal{Y}$, and $\pi_\theta(\cdot \mid x)$ is a $C^3$ parametric family with log-density $\log \pi_\theta$ locally smooth. Expectations without explicit subscript are conditional on $x$.
We first show useful lemmas here.

**Lemma E.1** (Frozen-policy surrogate is second-order tight). *Let $f(\theta) := \mathcal{L}_{\mathrm{PG}}(\theta)$ in equation 4 and $g(\theta) := \widehat{\mathcal{L}}_{\mathrm{PG}}(\theta; \theta_t)$ be the frozen-policy surrogate with $A_{\theta_t}$. Then $f(\theta_t) = g(\theta_t)$ and $\nabla f(\theta_t) = \nabla g(\theta_t)$. If $\nabla f$ and $\nabla g$ are $L$-Lipschitz in a neighborhood of $\theta_t$, then*

$$\left| f(\theta_t + \Delta\theta) - g(\theta_t + \Delta\theta) \right| \leq \frac{L}{2} \|\Delta\theta\|^2.$$

*Proof.* At $\theta_t$, both objectives evaluate to $-\mathbb{E}_{\pi_{\theta_t}}[A_{\theta_t} \log \pi_{\theta_t}]$. For the gradient, using the log-derivative trick and the centering of $A_{\theta_t}$, both yield $-\mathbb{E}_{\pi_{\theta_t}}[A_{\theta_t} \nabla \log \pi_{\theta_t}]$. Thus $f(\theta_t) = g(\theta_t)$ and $\nabla f(\theta_t) = \nabla g(\theta_t)$. The bound is the standard second-order Taylor remainder under Lipschitz gradients. □

### 1: Exponential tilting and M-projection

**Lemma E.2** (Gibbs variational principle / exponential tilting). *Fix $\beta > 0$ and a full-support reference $q(\cdot \mid x)$. Then*

$$\max_{\pi \ll q} \left\{ \mathbb{E}_{y \sim \pi}[R(x, y)] - \beta D_{\mathrm{KL}}(\pi \| q) \right\}$$

*is uniquely maximized by*

$$\tilde{q}_\beta(y \mid x) = \frac{q(y \mid x) \exp(R(x, y)/\beta)}{\mathbb{E}_{y \sim q}[\exp(R(x, y)/\beta)]}.$$

*Proof.* Consider $\mathcal{L}(\pi, \lambda) = \mathbb{E}_\pi[R] - \beta \mathbb{E}_\pi[\log \frac{\pi}{q}] + \lambda(\sum_y \pi(y) - 1)$. Stationarity in $\pi$ gives $\log \frac{\pi}{q} = R/\beta - \lambda - 1$, hence $\pi \propto q\, e^{R/\beta}$. Strict concavity in $\pi$ yields uniqueness. □

**Lemma E.3** (Policy Gradient Update as Parametric $M$-projection). *For fixed $\tilde{q}_\beta$,*

$$\arg\min_\theta D_{\mathrm{KL}}(\tilde{q}_\beta \| \pi_\theta) = \arg\max_\theta \mathbb{E}_{y \sim \tilde{q}_\beta}[\log \pi_\theta(y \mid x)].$$

*Proof.* $D_{\mathrm{KL}}(\tilde{q}_\beta \| \pi_\theta) = \mathbb{E}_{\tilde{q}_\beta}[\log \tilde{q}_\beta] - \mathbb{E}_{\tilde{q}_\beta}[\log \pi_\theta]$, where the first term is $\theta$-independent. We omit the full proof here, with one can be found in Shenfeld et al. (2025). □

### 2: Local second-order identities

**Lemma E.4** (Local Pythagorean identity for the $M$-projection). *Let $f(\theta) := D_{\mathrm{KL}}(\tilde{q}_\beta \| \pi_\theta) = \mathbb{E}_{\tilde{q}_\beta}[-\log \pi_\theta] + \text{const}$. Assume $\log \pi_\theta$ is $C^3$ near $\theta$, and let $\theta^+ \in \arg\min f$. Writing $\Delta := \theta^+ - \theta$, for $\|\Delta\|$ small,*

$$f(\theta) - f(\theta^+) = \tfrac{1}{2} \Delta^\top H_{\tilde{q}}(\theta)\, \Delta + O(\|\Delta\|^3), \quad H_{\tilde{q}}(\theta) := -\mathbb{E}_{\tilde{q}_\beta}[\nabla^2 \log \pi_\theta].$$

*Proof.* Taylor-expand $f$ at $\theta^+$: $f(\theta) = f(\theta^+) + \tfrac{1}{2}\Delta^\top H_{\tilde{q}}(\theta^+)\Delta + O(\|\Delta\|^3)$ since $\nabla f(\theta^+) = 0$. Local $C^3$ smoothness implies $H_{\tilde{q}}(\theta^+) = H_{\tilde{q}}(\theta) + O(\|\Delta\|)$, which is absorbed into the cubic remainder. □

**Lemma E.5** (Quadratic expansion of policy KL). *Let $F(\theta) := -\mathbb{E}_{\pi_\theta}[\nabla^2 \log \pi_\theta]$ be the Fisher information. Then*

$$D_{\mathrm{KL}}(\pi_{\theta+\Delta} \| \pi_\theta) = \tfrac{1}{2} \Delta^\top F(\theta)\, \Delta + O(\|\Delta\|^3).$$

*Proof.* Expand $\log \frac{\pi_{\theta+\Delta}}{\pi_\theta} = \Delta^\top \nabla \log \pi_\theta + \tfrac{1}{2}\Delta^\top \nabla^2 \log \pi_\theta\, \Delta + O(\|\Delta\|^3)$, take expectation under $\pi_{\theta+\Delta} = \pi_\theta + O(\|\Delta\|)$, use $\mathbb{E}_{\pi_\theta}[\nabla \log \pi_\theta] = 0$ and $-\mathbb{E}_{\pi_\theta}[\nabla^2 \log \pi_\theta] = F(\theta)$. □

### 3. Relating projection Hessian and Fisher under small tilt

**Lemma E.6** (Hessian–Fisher proximity). *Suppose $\|\nabla^2 \log \pi_\theta(y \mid x)\|_{\mathrm{op}} \leq L$ uniformly near $\theta$. Then*

$$\left\| H_{\tilde{q}}(\theta) - F(\theta) \right\|_{\mathrm{op}} \leq 2L\, \mathrm{TV}(\tilde{q}_\beta, \pi_\theta) \leq L\sqrt{2 D_{\mathrm{KL}}(\tilde{q}_\beta \| \pi_\theta)}.$$

*In particular, with $\kappa := D_{\mathrm{KL}}(\tilde{q}_\beta \| \pi_\theta) \to 0$, we have $H_{\tilde{q}}(\theta) = (1 + O(\sqrt{\kappa}))\, F(\theta)$ as quadratic forms.*

*Proof.* For bounded matrix-valued $h$, $\|\mathbb{E}_{\tilde{q}} h - \mathbb{E}_\pi h\|_{\mathrm{op}} \leq 2\|h\|_\infty \mathrm{TV}(\tilde{q}, \pi)$. Apply this with $h := -\nabla^2 \log \pi_\theta$ and Pinsker's inequality $\mathrm{TV}(p, q) \leq \sqrt{\tfrac{1}{2} D_{\mathrm{KL}}(p \| q)}$. □

**4. Remainder control**

**Lemma E.7** (Cubic remainder is $o(f)$). *If $H_{\tilde{q}}(\theta) \succeq mI$ on the update subspace (local strong convexity), then for $\|\Delta\|$ small*

$$\|\Delta\|^2 \leq \tfrac{2}{m}\left(f(\theta) - f(\theta^+)\right), \qquad O(\|\Delta\|^3) = o\!\left(f(\theta)\right).$$

*Proof.* From Lemma E.4, $f(\theta) - f(\theta^+) \geq \tfrac{m}{2}\|\Delta\|^2 + O(\|\Delta\|^3)$. Rearranging yields $\|\Delta\|^2 = O(f(\theta) - f(\theta^+))$, so the cubic term is lower order. $\qquad\square$

### E.2.1    PROOF OF PROPOSITION 3.1

*Proof of Proposition 3.1.* Let $f(\theta) = D_{\mathrm{KL}}(\tilde{q}_\beta \| \pi_\theta)$ and $\Delta = \theta^+ - \theta$. By Lemma E.4,

$$f(\theta) - f(\theta^+) = \tfrac{1}{2}\,\Delta^\top H_{\tilde{q}}(\theta)\Delta + O(\|\Delta\|^3).$$

By Lemma E.5,

$$D_{\mathrm{KL}}(\pi_{\theta^+} \| \pi_\theta) = \tfrac{1}{2}\,\Delta^\top F(\theta)\Delta + O(\|\Delta\|^3).$$

By Lemma E.6 with $\kappa = f(\theta)$, $\Delta^\top F\Delta = \left(1 + O(\sqrt{\kappa})\right)\Delta^\top H_{\tilde{q}}\Delta$. Hence

$$D_{\mathrm{KL}}(\pi_{\theta^+} \| \pi_\theta) = \left(1 + O(\sqrt{\kappa})\right)\left(f(\theta) - f(\theta^+)\right) \;+\; O(\|\Delta\|^3).$$

Since $f(\theta^+) \geq 0$, $f(\theta) - f(\theta^+) \leq f(\theta) = \kappa$. By Lemma E.7, $O(\|\Delta\|^3) = o(f(\theta))$. Therefore

$$D_{\mathrm{KL}}(\pi_{\theta^+} \| \pi_\theta) \;\leq\; (1 + o(1))\, f(\theta) \;=\; (1 + o(1))\, D_{\mathrm{KL}}(\tilde{q}_\beta \| \pi_\theta),$$

which is the desired inequality. $\qquad\square$

### E.2.2    PROOF OF PROPOSITION 3.2

*Proof of Proposition 3.2.* By the quadratic expansion of policy KL (Lemma E.5),

$$D_{\mathrm{KL}}(\pi_{\theta+\Delta} \| \pi_\theta) = \tfrac{1}{2}\,\Delta^\top F(\theta)\Delta \;+\; R(\Delta), \qquad |R(\Delta)| \leq C\|\Delta\|^3 \tag{12}$$

for some local constant $C > 0$ (from $C^3$ smoothness). Let $a := \Delta^\top F(\theta)\Delta$. Using the spectral lower bound $F(\theta) \succeq \mu I$ on the update subspace,

$$\|\Delta\|^2 \leq \tfrac{a}{\mu}. \tag{13}$$

Combining equation 12–equation 13 yields

$$D_{\mathrm{KL}}(\pi_{\theta+\Delta} \| \pi_\theta) \;\geq\; \tfrac{1}{2}\,a \;-\; C\left(\tfrac{a}{\mu}\right)^{3/2}.$$

Since $D_{\mathrm{KL}}(\pi_{\theta^+} \| \pi_\theta) \leq K$, we have

$$K \;\geq\; \tfrac{1}{2}\,a \;-\; C\,\mu^{-3/2} a^{3/2}. \tag{14}$$

For $a$ sufficiently small (equivalently, $K$ small), the cubic term is dominated by the linear term: choose $a_0 > 0$ so that $C\,\mu^{-3/2}\sqrt{a} \leq \tfrac{1}{4}$ whenever $0 < a \leq a_0$. Then from equation 14

$$K \;\geq\; \left(\tfrac{1}{2} - \tfrac{1}{4}\right)a \;=\; \tfrac{1}{4}\,a \quad \Rightarrow \quad a \;\leq\; 4K.$$

Substituting $a \leq 4K$ back into equation 12 refines the remainder: $|R(\Delta)| \leq C\|\Delta\|^3 \leq C(a/\mu)^{3/2} = O(K^{3/2}) = o(K)$, so $D_{\mathrm{KL}}(\pi_{\theta+\Delta} \| \pi_\theta) = \tfrac{1}{2}a + o(K)$. Hence $a = 2\,D_{\mathrm{KL}}(\pi_{\theta+\Delta} \| \pi_\theta) + o(K) \leq 2K + o(K)$, i.e.

$$\Delta^\top F(\theta)\Delta \;\leq\; 2K\,(1 + o(1)).$$

Taking square roots gives the Fisher-norm bound in equation 6: $\|\Delta\|_{F(\theta)} = \sqrt{\Delta^\top F(\theta)\Delta} \leq \sqrt{2K}\,(1 + o(1))$. The Euclidean bound follows from equation 13:

$$\|\Delta\|_2 \;\leq\; \sqrt{\tfrac{\Delta^\top F(\theta)\Delta}{\mu}} \;\leq\; \sqrt{\tfrac{2K}{\mu}}\,(1 + o(1)).$$

Finally, for any parameter block $W \subset \theta$, its Frobenius change is the $\ell_2$-norm of the corresponding subvector of $\Delta$; therefore $\|\Delta W\|_F \leq \|\Delta\|_2$. $\qquad\square$

### E.2.3  ONE-STEP KL BUDGET (USED IN GATE II)

**Corollary E.8** (KL budget). *If $D_{\mathrm{KL}}(\pi_{\theta^+}\|\pi_\theta) \le K$, then*

$$\tfrac{1}{2}\,\Delta^\top F(\theta)\,\Delta \;\le\; K\,(1 + o(1)).$$

*Proof.* Apply Lemma E.5 and Lemma E.7. □

### E.2.4  TRUST-REGION / CLIPPING BOUND (FOR $\beta = 0$)

**Lemma E.9** (Implicit KL leash from ratio clipping). *Let $r_t = \frac{\pi_{\theta^+}(y_t|x,y_{<t})}{\pi_\theta(y_t|x,y_{<t})}$ and suppose clipping enforces $r_t \in [1 - \varepsilon,\, 1 + \varepsilon]$ on the batch. Then*

$$\widehat{D}_{\mathrm{KL}}(\pi_{\theta^+}\|\pi_\theta) \;\le\; \widehat{\mathbb{E}}[T(x)] \cdot \max\{-\log(1-\varepsilon),\, \log(1+\varepsilon)\} \;=\; O(\varepsilon) \cdot \widehat{\mathbb{E}}[T(x)],$$

*and in the small-step regime (mean-zero advantage) this tightens to $O(\varepsilon^2)$.*

*Proof.* Autoregressive factorization gives $D_{\mathrm{KL}}(\pi_{\theta^+}\|\pi_\theta) = \mathbb{E}_{\pi_{\theta^+}}[\sum_t \log r_t]$. Because $\log r_t \in [\log(1-\varepsilon), \log(1+\varepsilon)]$, we have $|\log r_t| \le c(\varepsilon)$; summing over $t$ and taking batch expectation yields the stated bound. Using $\log(1 \pm \varepsilon) = \pm\varepsilon + O(\varepsilon^2)$ and small-step arguments gives $O(\varepsilon^2)$. □

### E.3  PROOFS FOR GATE II (SEC. 3.2)

**Setup (layer-conditioned budget).**  Partition $\theta = (\mathrm{vec}(W), \theta_{\neg W})$ and let the Fisher at $\theta = \theta_t$ be

$$F(\theta) = \begin{bmatrix} F_{W,W} & F_{W,\neg W} \\ F_{\neg W,W} & F_{\neg W,\neg W} \end{bmatrix} \succeq 0.$$

For a one-step update $\Delta\theta$, the global KL leash implies $\tfrac{1}{2}\Delta\theta^\top F(\theta)\Delta\theta \le K$. Define the layer-conditioned curvature

$$S_W \coloneqq F_{W,W} - F_{W,\neg W} F_{\neg W,\neg W}^{-1} F_{\neg W,W} \succeq 0,$$

and the per-layer budget $\delta_W \coloneqq \tfrac{1}{2}\mathrm{vec}(\Delta W)^\top S_W \mathrm{vec}(\Delta W) \le K$. Let $\mu_W \coloneqq \lambda_{\min}(S_W) > 0$ on the update subspace.

**Lemma E.10** (Layer-conditioned Frobenius/operator bounds). $\|\Delta W\|_F \le \sqrt{2\delta_W/\mu_W}$ *and* $\|\Delta W\|_2 \le \|\Delta W\|_F$.

*Proof.* Since $S_W \succeq \mu_W I$, $\delta_W \ge \tfrac{1}{2}\mu_W\|\Delta W\|_F^2$. □ □

**Lemma E.11** (Wedin's sin–$\Theta$). *For $W_+ = W_0 + \Delta W$, the principal subspace angles satisfy* $\|\sin\Theta(U_k(W_0), U_k(W_+))\|_2 \le \|\Delta W\|_2/\gamma_k$ *and similarly for $V_k$.* □

**Lemma E.12** (Weyl/Mirsky and Hoffman–Wielandt). $|\sigma_k(W_+) - \sigma_k(W_0)| \le \|\Delta W\|_2$ *and* $\sum_i(\sigma_i(W_+) - \sigma_i(W_0))^2 \le \|\Delta W\|_F^2$. □

**Corollary E.13** (Projection stability). *With the same assumptions,*

$$\left\|U_k(W_0)U_k(W_0)^\top - U_k(W_+)U_k(W_+)^\top\right\|_2 \;=\; \left\|\sin\Theta\big(U_k(W_0), U_k(W_+)\big)\right\|_2 \;\le\; \frac{\sqrt{2\delta_W/\mu_W}}{\gamma_k}.$$

*The analogous bound holds for the right subspaces with $V_k$.* Interpretation. *The leading invariant subspaces rotate by at most $O\big(\sqrt{\delta_W/\mu_W}/\gamma_k\big)$; when the gap is moderate, the rotation is small.* □

# F  MORE VISUALIZATION

## F.1  JACCARD MATRIX

RL updates are highly consistent across independent training runs. Fig. 8 shows the pair-wise Jaccard similarity between the final update masks from five RLVR runs on different data and algorithms. The high similarity scores demonstrate that the optimization process consistently targets the same subset of parameters, providing strong evidence for a deterministic, non-random optimization bias.

## F.2  SPECTRUM SHIFT FOR DS-1.5B AND QWEN3-1

We also show the spectrum shift for DS-1.5B and Qwen3-1 here.

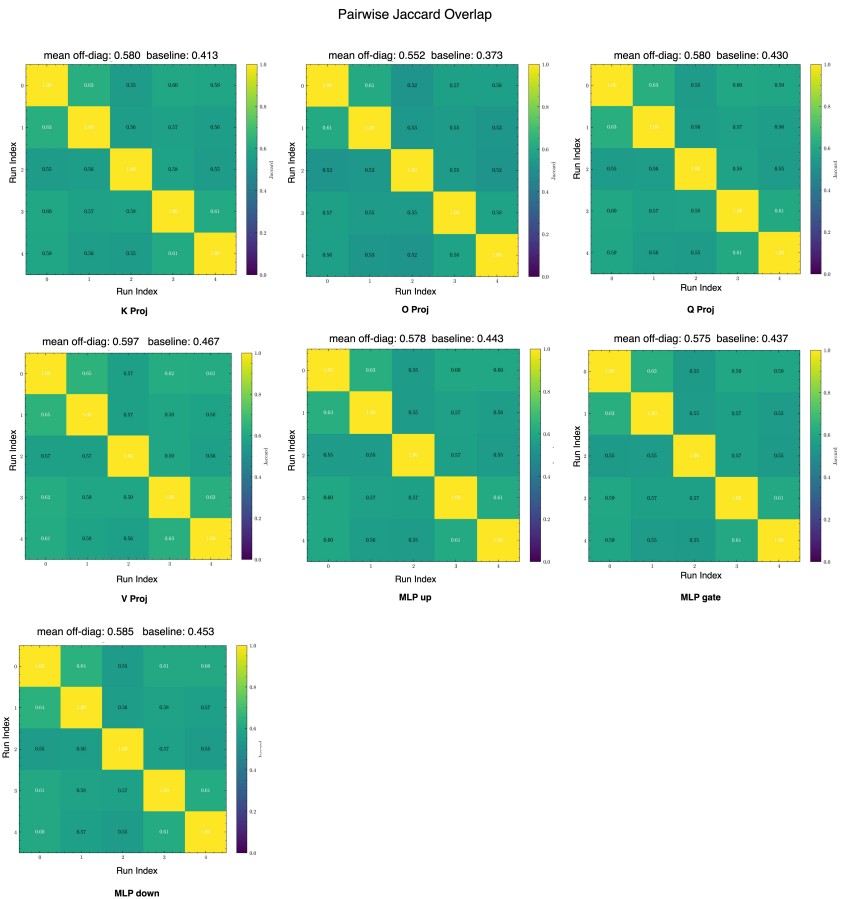

Figure 8: Pair-wise Jaccard similarity of update masks from five independent RLVR runs on Layer 13 of the `DS-Distill-Qwen-1.5B` model.

Table 3: Performance of DS-Qwen-1.5B with different masking strategies with a extended training window to 500 steps. Parameter counts shown are for linear layers only, excluding the embedding and head layers. Detailed evaluation settings are available in Appendix C.2. *We observe that training only on principal weights $M_{princ}$ results in a clear accuracy gap compared to both the dense baseline and its complement $M_{princ}^c$. The models using the $M_{low}$ and $M_{princ}^c \cup M_{lowest}$ masks achieve performance closest to the dense baseline.*

| Model | Mask | Math500 | AMC23 | AIME24 | AIME25 | Average | #params |
|-------|------|---------|-------|--------|--------|---------|---------|
| | Dense | **84.5** | **83.52** | 38.28 | **28.075** | **58.59** | 100% |
| | $M_{princ}$ | 83.60 | 78.83 | 34.06 | 25.63 | 55.44 | 50% |
| | $M_{princ}^c$ | 84.0 | 77.97 | 38.64 | 27.81 | 56.90 | 50% |
| DS-Qwen-1.5B | $M_{low}$ | 83.8 | 82.42 | 37.03 | 27.82 | 57.77 | 58.59% |
| | $M_{princ}^c \cup M_{low}$ | 84.10 | 81.41 | **40.30** | 27.70 | 58.37 | 74.02% |
| | Random-$M_{princ}^c \cup M_{low}$ | 84.10 | 81.72 | 34.69 | 27.34 | 56.89 | 74.02% |

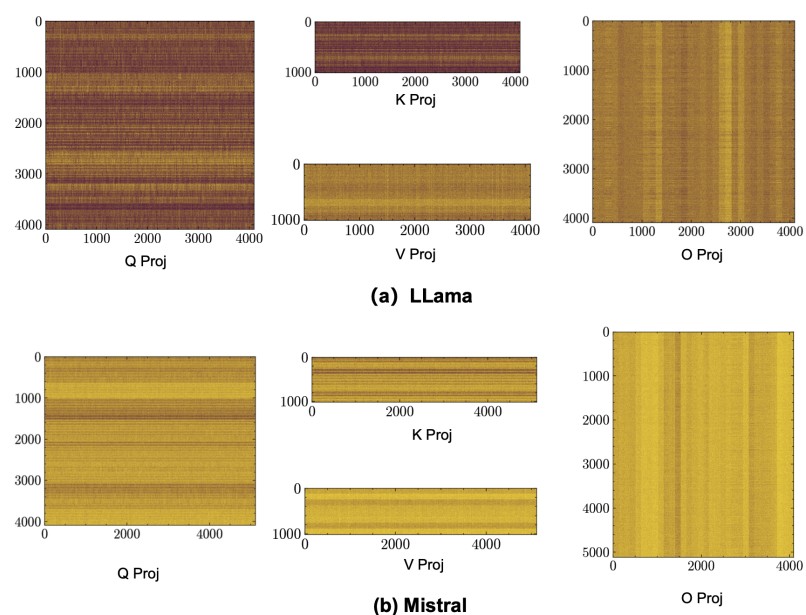

Figure 9: Structured Update observed on LLama and mistral models.

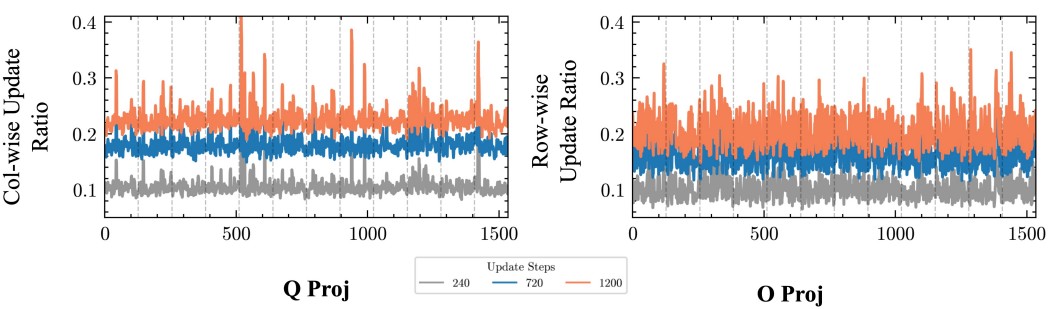

Figure 10: Temporal emergence of the optimization bias with row and column-wise update ratios for the 13th attention block across gradient update steps ($t \in \{240, 720, 1200\}$), smoothed with a 3-step window. *The column-wise (Q) and row-wise (O) update ratios show a much weaker bias.*

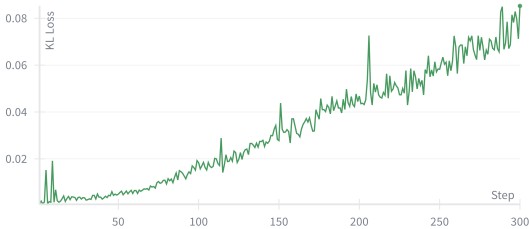

Figure 11: Token-wise KL loss. We show the token-wise KL loss during a DAPO run without a KL loss penalty, which shows a steadily increasing KL loss instead of being unconstrained.

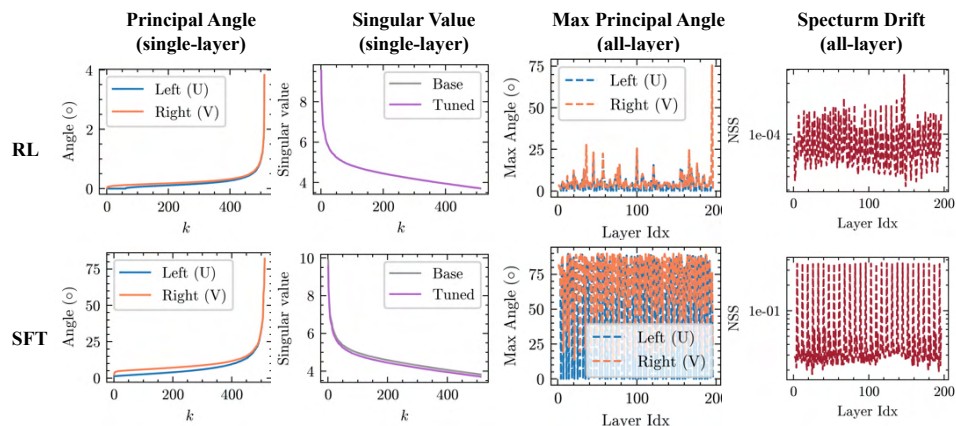

Figure 12: The spectrum probe results on the RL and SFT version on the `DS-Distill-Qwen-1.5B` Liu et al. (2025a). RLVR shows surprisingly stable top-k spectrum with minimal subspace rotation and top-k eigenvalue changes.

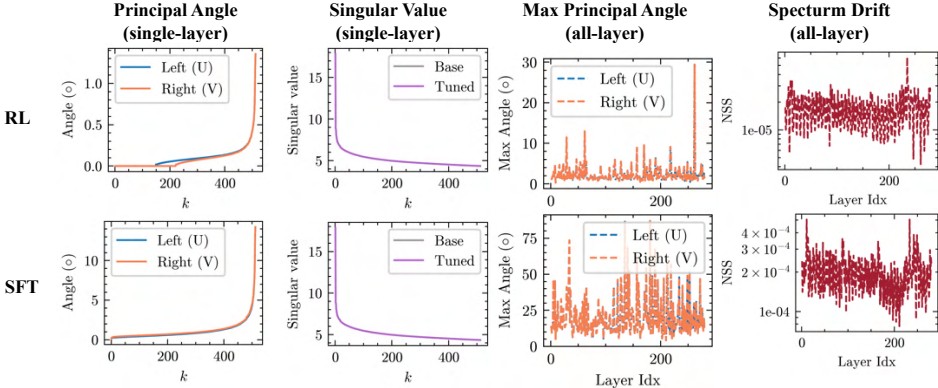

Figure 13: The spectrum probe results on the RL and SFT version on the Qwen3-14B Huan et al. (2025). RLVR shows surprisingly stable top-k spectrum with minimal subspace rotation and top-k eigenvalue changes.

## G ADDED RESULTS DURING REBUTTAL

### G.1 CONSENSUS RATIO AND ALGORITHM ROBUSTNESS

To further validate our findings on the "Implicit Compass," we extended our analysis to **Llama-3.2-3b-Instruct** across five distinct reinforcement learning configurations. We varied the algorithm (Majority-Voting, Self-Certainty, Co-rewarding, GRPO) and the dataset (MATH vs. DAPO-14k). The resulting sparsity levels are detailed in Table 4.

Table 4: **Additional Rebuttal Runs (Llama-3.2-3b-Instruct).** All from Co-reward Comparison of update sparsity across different RL algorithms and datasets.

| Dataset | Method | Sparsity |
|---------|--------|----------|
| MATH | Majority-Voting | 71.21% |
| MATH | Self-Certainty | 83.24% |
| MATH | Co-rewarding | 71.86% |
| MATH | GRPO | 71.28% |
| DAPO-14k | GRPO | 66.00% |

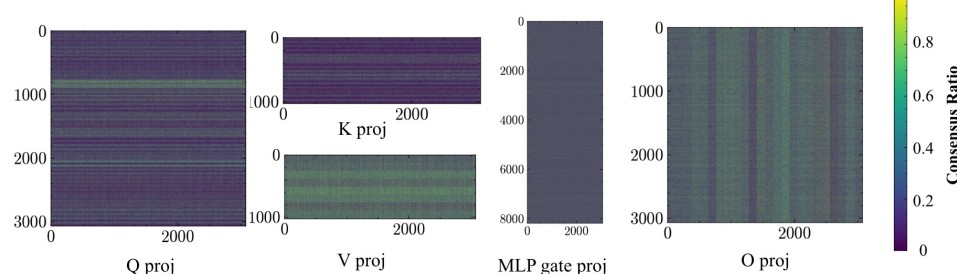

Figure 14: **Robustness of Consensus across Algorithms.** Visualization of the update consensus masks for Llama-3.2-3b-Instruct across five different runs using distinct RL algorithms (Majority-Voting, Self-Certainty, Co-rewarding, GRPO) and datasets (MATH, DAPO-14k). The consistent observation of "stripe-like" structures (row-wise in Q/K/V, column-wise in O) confirms that the "Implicit Compass" is intrinsic to the model's optimization dynamics and robust to the choice of reinforcement learning method.

**Same Model, Different Algorithms (Consensus Check):** We applied the consensus ratio metric $C_{\ell,ij}$ to these new runs. As shown in Figure 9 (Appendix), we observe the identical "stripe-like" consensus in update masks. Even without changing the task, the fact that independent RL runs (using disjoint data and variants like Self-Certainty or Co-rewarding) consistently route updates to the same regions confirms the "Implicit Compass" is intrinsic to the optimization dynamics, not an artifact of a specific model.

### G.2 NEW CHECK ON AGENT, EMBODIED AND RLHF TASKS

To assess the universality of the RLVR optimization signatures—specifically the "Implicit Compass" and "Three-Gate" dynamics—we extended our analysis beyond mathematical reasoning to a broader suite of post-training paradigms. These include multi-turn agentic workflows, tool-augmented reasoning, and standard preference optimization (RLHF).

**Model Details.** The specific checkpoints analyzed, along with their corresponding base models and task domains (ranging from web navigation to robotic control), are detailed in Table 5.

**Spectral Stability.** We first examine the spectral properties of the weight matrices after RL fine-tuning. Across agents (Figure 15), embodied AI models (Figure 17), and RLHF checkpoints (Figure 16), we observe a striking consistency with our reasoning-task findings. Specifically, the layer spectra remain stable, exhibiting near-identical singular values between the RL-finetuned weights ($W_{\text{RL}}$) and their base counterparts ($W_{\text{Base}}$). Furthermore, the principal subspaces (top-$k$ singular vectors) undergo only small rotation.

Table 5: **Model List for analyzed checkpoints for agentic and embodied AI (robotics manipulation) tasks and RLHF algorithms.**

| Category | Base Model | FT Model | Algorithm | Data | Sparsity |
|---|---|---|---|---|---|
| Agent | Qwen3-8B | SkyRL-Agent-WebResearch-8B | GRPO | WebResearch | 40.56% |
| | Qwen3-8B | VT-deepsearch-8B | GRPO | Deepsearch | 89.67% |
| | Qwen3-8B | VT-SWE-8B | GRPO | SWE | 84.32% |
| | Qwen2.5-7B-Instruct | agentflow-planner-7b | Flow-GRPO | Planning | 80.99% |
| | Qwen2.5-7B-Instruct | GiGPO-Qwen2.5-7B-Instruct-WebShop | GiGPO | WebShop | 51.7% |
| | Qwen2.5-7B-Instruct | GiGPO-Qwen2.5-7B-Instruct-ALFWorld | GiGPO | ALFWorld | 62.08% |
| RLHF | Meta-Llama-3-8B-Instruct | Llama-3-Instruct-8B-DPO | DPO | instruction-following | 82.38% |
| | Meta-Llama-3-8B-Instruct | Llama-3-Instruct-8B-SimPO | SimPO | instruction-following | 71.00% |
| Embodied AI | openvla/openvla-7b | Openvla-oft-SFT-libero10-trajall | SFT | Robotic manipulation Liu et al. (2023) | 3.46% |
| | Openvla-oft-SFT-libero10-traj1 | openvla-oft-libero10-traj1-rl | GRPO | Robotic manipulation Liu et al. (2023) | 35.04% |
| | Qwen2.5-VL-3B-Instruct | Embodied-R1-3B-v1 | GRPO | Robotic Manipulation | 44.28% |

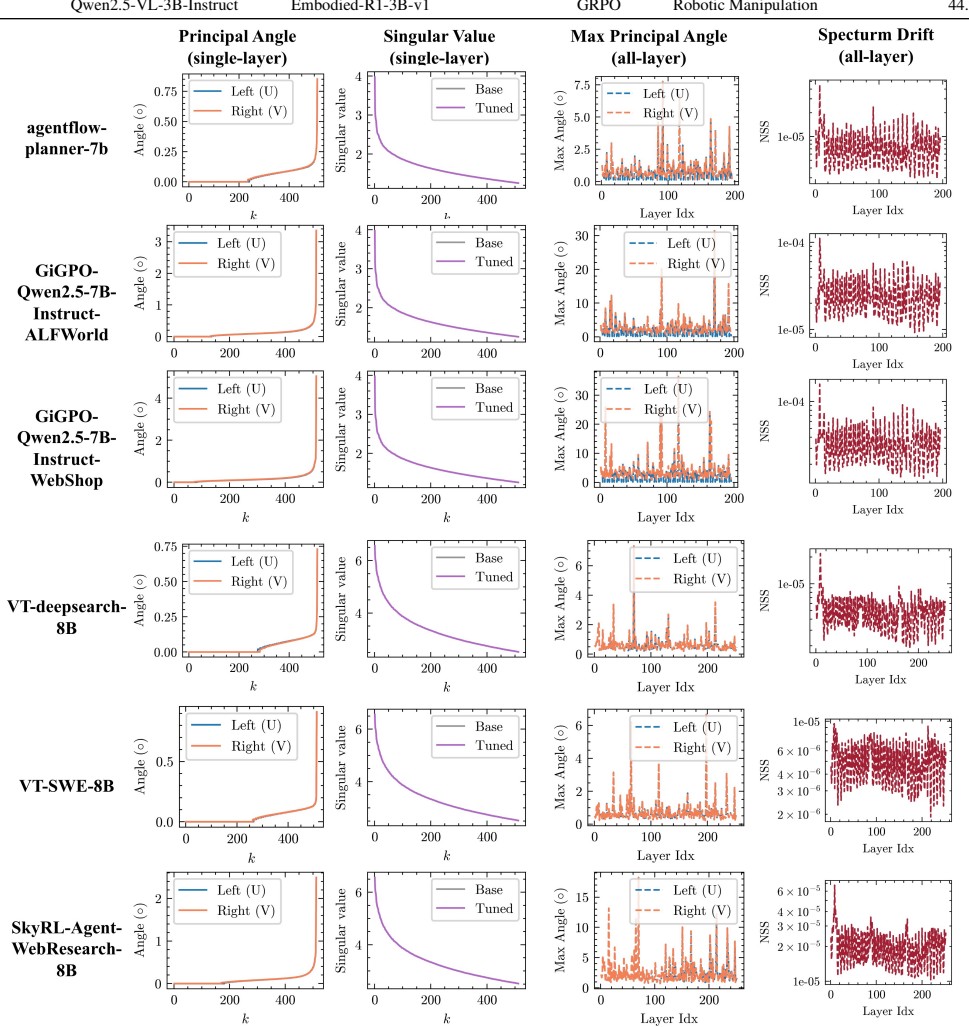

Figure 15: **Spectrum under RL in agent tasks.** In agent settings, including multi-turn interactions and tool use, RL leaves layer singular-value spectra nearly unchanged and induces only small rotations of the top-$k$ singular subspaces, consistent with the spectrum-preserving, off-principal RLVR regime. Results for RL with human feedback (RLHF), which exhibit the same optimization signature, appear in Fig. 16. For consistency, we use the *second block O*-projection layer as an exemplar single-layer readout.

**Off-Principal Routing.** Finally, we verify the spatial distribution of the updates. Figure 18 visualizes the update masks $(M_\ell)$ against the principal masks $(P_\ell^{(k)})$ for representative agent and embodied models. Consistent with the "Implicit Compass" hypothesis, the updates strictly avoid the principla parts.

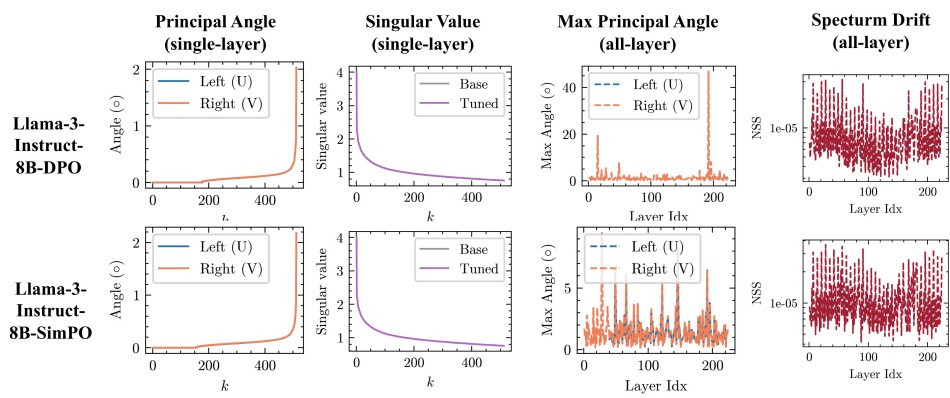

Figure 16: **Spectral geometry under RLHF setting Meng et al. (2024b).** Across RLHF checkpoints, RL training preserves layer spectra and induces only minor rotation of the top-$k$ subspaces, consistent with the RLVR regime.

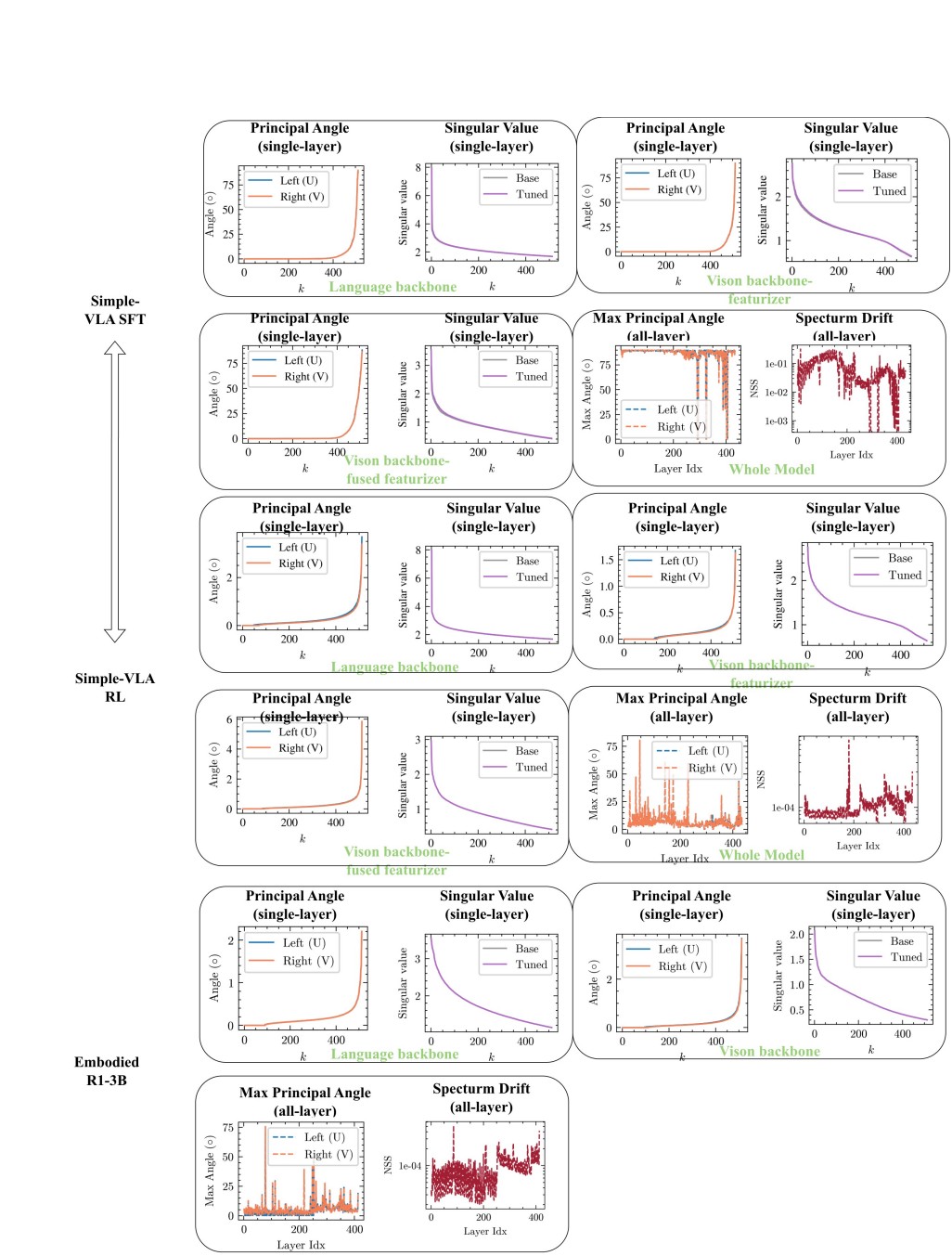

Figure 17: **Spectral geometry under embodied AI tasks: SimpleVLA-RL (Li et al., 2025a) and Embodied R1 (Yuan et al., 2025).** Across language and vision backbone, RL training preserves layer spectra and induces only minor rotation of the top-$k$ subspaces.

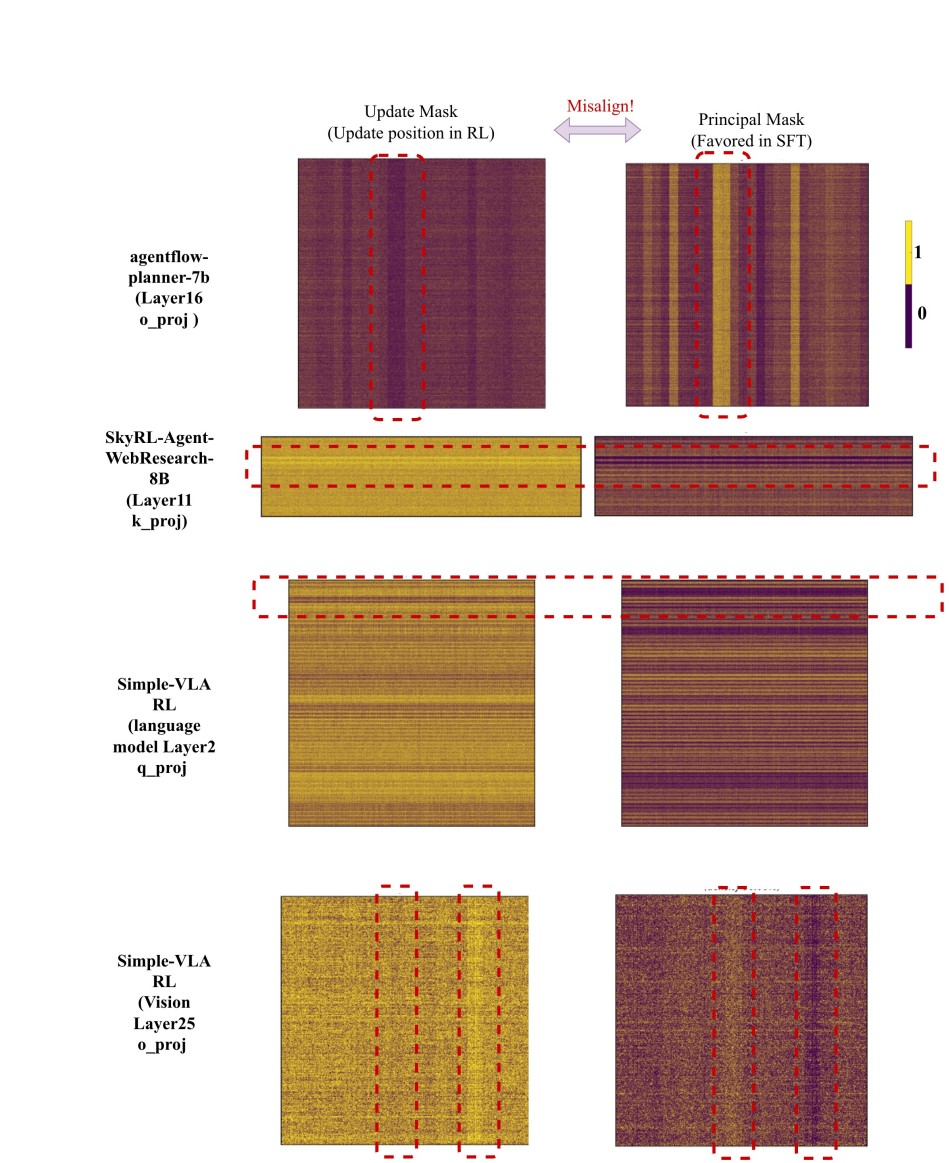

Figure 18: **Update–principal misalignment in RL-trained agents and embodied AI models.** The figure visualizes the bf16-aware *update mask* $M_\ell$ (left, showing locations changed under RL) versus the *principal mask* $P_\ell^{(k)}$ (right, showing top-$k$ singular-subspace support) for representative layers across different domains. Top row: AGENTFLOW-PLANNER-7B (Layer 16, $o_{\mathrm{proj}}$). Second row: SKYRL-AGENT-WEBRESEARCH-8B (Layer 11, $k_{\mathrm{proj}}$). Third row: SIMPLE-VLA-RL (language model Layer 2, $q_{\mathrm{proj}}$). Bottom row: SIMPLE-VLA-RL (Vision Layer 25, $o_{\mathrm{proj}}$). Dashed blue boxes highlight regions where RL updates concentrate *outside* principal-weight bands. This consistently shows that RL updates are misaligned with the principal subspace, indicating a robust off-principal routing mechanism across agent, tool-use, and embodied AI settings.

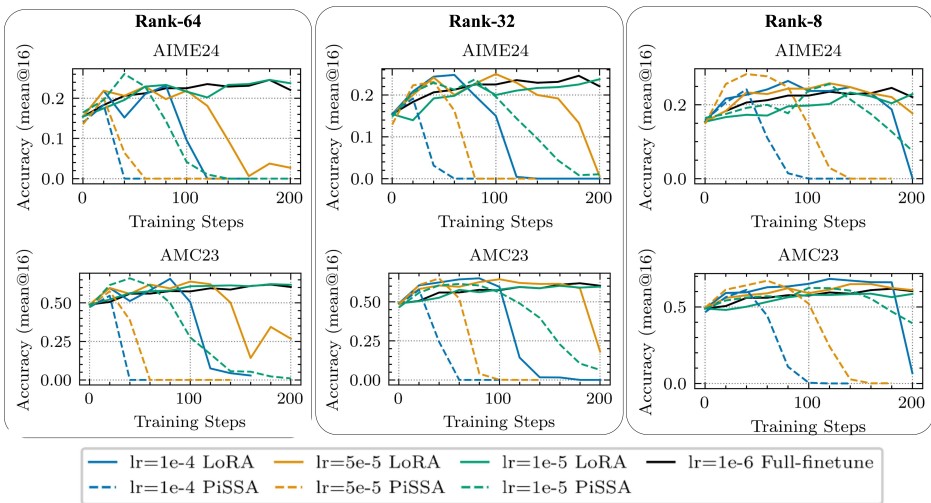

Figure 19: **LoRA vs. PiSSA on `DS-Qwen-1.5B` (DeepMath-103K).** We sweep ranks $\{8, 32, 64\}$ and learning rates $\{1\times10^{-4}, 5\times10^{-5}, 1\times10^{-5}\}$ for 200 steps, reporting pass@1 (avg@16) on AIME24 (top) and AMC23 (bottom). Across settings, **PiSSA** (principal-targeted) provides *no additional gains* over LoRA and, at higher learning rates that force principal-direction updates, *often collapses early*; LoRA remains more stable. This supports our geometric account: forcing updates into principal directions (favored in SFT) is misaligned with RL, offering no obvious gain and leading to training collapse when scaling up learning rates.

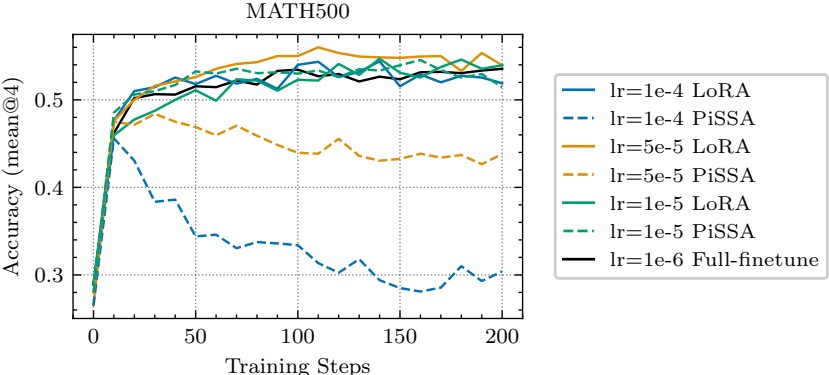

Figure 20: **LoRA vs. PiSSA on `LLaMA-3.2-3B`.** We sweep learning rates $\{1\times10^{-4}, 5\times10^{-5}, 1\times10^{-5}\}$ with a fixed rank of 64 for 200 steps, reporting pass@1 (mean@4) on MATH500. Consistent with the `DS-Qwen-1.5B` results in Fig. 19, **PiSSA** provides *no additional gain* over LoRA and, under higher learning rates that emphasize principal-direction updates, *often collapses early*.

