# OpenReview forum: "Why RL Updates Look Sparse: An Implicit Compass Drives Optimization Bias"
_ICLR.cc/2026/Conference — Submitted to ICLR 2026_

### Official Review · Reviewer_QWqF · 2025-10-26

**Soundness:** 2
**Presentation:** 2
**Contribution:** 1
**Rating:** 2
**Confidence:** 3

**Summary:**

The paper investigates why reinforcement learning (RL) fine-tuning of large language models appears to modify only a small fraction of parameters while producing large changes to the end-to-end map.
It proposes that this apparent sparsity reflects an underlying optimization bias, the implicit compass, that steers updates toward certain regions of the model parameter space.
To explain this, the authors introduce a Three-Gate Theory: (1) a KL-regularization “leash”, (2) the pretrained model’s geometry directs those constrained updates, and (3) limited bfloat16 precision amplifies the effect by hiding some parts of updates.
Empirical analyses across several RL variants and models show stable, structured update patterns, and minimal overlap with principal weights, supporting the theory.
The paper concludes that RL’s distinctive optimization dynamics differ from supervised fine-tuning and that understanding this bias could inform more efficient RL fine-tuning methods.

**Strengths:**

To explain the success of RL in LLM finetuning is a timely topic. The main theoretical claims are well substantiated for the first two gates. The work discusses an important design principle for finetuning in general.

**Weaknesses:**

1) The first two gates are related so they are not actually two distinct phenomena. The KL "leash" is causing the minimal change in parameters.

2) Furthermore, the fact that the KL divergence update leads to minimal updates is not surprising as it is a well known result understood in optimization literature as the optimization geometry [5]. For example in RL the GRPO can be written as an exponential update as in Theorem 1 [6] which leads to sparse updates.

3) The third gate is not well motivated by experimental evidence as there are only BF16 results in Table 1. First of all would this not hold for all methods i.e. less precision would act as rounding therefore less updates.

4) While the work acknowledges [1], it seems to provide similar insights than it. The subnetwork discovery in [1] is a stronger claim than the implicit regularization caused by the KL.

5) The work seems to use to much LLM generated text, evidence: The use of "Optimization Compass" (Line 351) instead of  Optimization induced regularization or implicit regularization this happens at many instances in the manuscript, sub- and super-random (Line 087) instead of positive or negative correlated.

6) There is large body of work on implicit regularization by optimization which seems to be quite important as the work is about an "implicit compass" that guides the optimization [5].

7) As the work finds thar RL leads affecting less parameters, more relevant finetuning methods in literature on subgroup finetuning are [2,3,4]. How would RL type of methods compare to these type of methods?


[1] Mukherjee, Sagnik et al. “Reinforcement Learning Finetunes Small Subnetworks in Large Language Models.” ArXiv abs/2505.11711 (2025): n. pag.

[2] Modoranu, Ionut-Vlad et al. “MicroAdam: Accurate Adaptive Optimization with Low Space Overhead and Provable Convergence.” ArXiv abs/2405.15593 (2024): n. pag.
[3] Rios, Jesus et al. “Sparsity May Be All You Need: Sparse Random Parameter Adaptation.” ArXiv abs/2502.15975 (2025): n. pag.
[4] Zhou, Chao et al. “Pay Attention to Small Weights.” ArXiv abs/2506.21374 (2025): n. pag.

[5] Gunasekar, Suriya et al. “Characterizing Implicit Bias in Terms of Optimization Geometry.” ArXiv abs/1802.08246 (2018): n. pag.
[6] Mroueh, Youssef. “Reinforcement Learning with Verifiable Rewards: GRPO's Effective Loss, Dynamics, and Success Amplification.” ArXiv abs/2503.06639 (2025): n. pag.

**Questions:**

See weaknesses.

---

> ### Author Response · Authors · 2025-11-26
> **Response to Reviewer QWqF part1**
>
> We thank the reviewer for the detailed feedback and for recognizing the timeliness of understanding why RL succeeds in LLM fine-tuning. We believe several concerns stem from a mismatch in perceived scope (sparsity vs. structural learning bias), and we appreciate the opportunity to address it.
>
>
> ## Q.0 Structural learning bias vs. “just sparsity / just small regularized updates”
>
> > "KL divergence update leads to minimal updates is not surprising...The subnetwork discovery in [1] is a stronger claim than the implicit regularization caused by the KL."
>
> *We respectfully feel that the review views our work as primarily investigating the existence of "sparsity" or small KL-regularized updates, likely stemming from our original title, “Why RL Updates Look Sparse”.* We believe this reflects a fundamental misunderstanding of our scope.
>
> Our actual contribution is to characterize **a structural learning bias** that governs where RL updates land in parameter space and how they reshape weights (the true learning dynamics), rather than merely how big they are.
>
>
> **(i) Magnitude vs. Direction (core distinction).** We fully agree that if our finding were solely "RL updates $||\Delta \theta||$ are constrained," our Gate I would fully account for this.
>
> * **Direction (Our Novelty):** The constraint on norm is "blind" to location; it does not explain why updates consistently land in specific regions of the same model across disjoint datasets and RL algorithms. In constrast, we show that RLVR updates are not random but are consistently steered into **off-principal regions** while preserving the core spectra. In other words, we explain the direction/location of updates (where the model chooses to move), not just their size.
>
>
> **(ii) Depth (Beyond Visual Sparsity):** Sparsity is merely a visual "readout" that varies with precision. The phenomenon of interest is the structural, spectrum-preserving, off-principal update pattern, a data- and algorithm-invariant mechanism. BF16 sparsity is simply how this pattern becomes visible under finite precision.
>
> * **New Evidence**: We added a precision sweep (BF16 $\to$ FP16 $\to$ FP32; see Table R1 in Response to Reviewer jERv). While visual sparsity diminishes in FP32, the **structural bias** (near-identical singular values and minimal rotation) persists. This proves the true mechanism is invariant to precision.
>
> **(iii) Practical Takeaway (Predictive Design)**: Truly understanding the underlying dynamics allows us to provide actionable insights. Unlike [1], which finds sparse masks after training, our theory allows us to predict the update region a priori (the "Safe Mask" in Sec 4.4). Most importantly, it guides geometry-aware algorithms, predicting the failure of PiSSA[10] (which targets principal weights) in RL (a open question from the recent thinking machine's blog [9])
>
> **(iv) Title change to de-emphasize sparsity.** As stated in Section 2.3, our goal is not to study sparsity, which is a visual artifact of the underlying learning bias. To avoid further misimpression that the paper is "only about sparsity," we will change the title in the revision to:
>
> > “The Path Not Taken: RLVR Provably Learns Off the Principals.”
>
> This better reflects that our main object of study is the structural learning dynamics of RLVR, with sparsity used only as a diagnostic lens.
>
> We hope this clarifies that our work is not merely a quantification of sparsity, but the first mechanistic theory of the Structural Learning Bias in RL, answering both where the bias comes from (Three-Gate Theory) and where it steers the model (Off-Principal Dynamics), with practical takeaways on designing RL-aware algorithms.

---

> ### Author Response · Authors · 2025-11-26
> **Response to Reviewer QWqF part2**
>
> ## Q.1 Distinguishing Gate I (Magnitude) from Gate II (Direction)
>
> > “The first two gates are related… The KL ‘leash’ is causing the minimal change.”
>
> We agree that KL-constrained RL leads to small updates, and this is exactly what our Gate I captures. However, Gate I and Gate II are about different aspects of the update:
>
> **(i) Constraint vs. steering.**
> * Gate I $\rightarrow$ Magnitude: The KL constraint limits the norm of the update ($||\Delta \theta||$). This explains why updates are small.
> * Gate II $\rightarrow$ Direction: The constraint on norm does not explain where the updates land. If Gate I were the sole factor, small updates would be randomly scattered noise. Gate II states that, given the pretrained geometry, RLVR systematically steers the KL-constrained updates into **off-principal** (away from the high-curvature) regions while preserving the core spectra.
>
> **(ii) Evidence that Gate I is not enough.** At the start of Sec. 3.2 (“From Gate I to location”), we explicitly formulated this distinction in the transition to Section 3.2 ("From Gate I to location").  Figure 2 provides the decisive evidence: we observe "stripe-like" consistency where independent runs on the same pretrained model target the exact same rows/columns. This deterministic spatial consensus confirms that the update is not merely "small" (Gate I) but actively "targeted" by the model's geometry (Gate II).
>
>
> ## Q.2  Theoretical Novelty vs. Optimization Literature [5, 6]
> > "KL divergence update leads to minimal updates is not surprising... understood in [5]... GRPOas in Theorem 1 [6] which leads to sparse updates.]."
>
> **(i) vs. Gunasekar et al. [5]:** This work characterizes implicit bias in linear models for supervised learning, under two families of losses: losses with a unique finite root and strict monotone losses. It does not predict the learning dynamics of Transformers (highly non-linear) or the specific spectral preservation we observe in RL.
>
> **(ii) vs. Mroueh [6]:** Theorem 1 in [6] describes an exponential update in Probability Space ($\pi_{new} \propto \pi_{old} e^R$). **Acually, we also use this lemma in Appendix E.2 as part of Gate I**, but this is a statement about the probability distribution, not the parameter update $||\Delta \theta||$; [6] does not derive a sparse or off-principal weight update pattern.
>
>
> ## Q.3 Precision is a lens, not the cause
> > "Gate III is not well motivated... First of all would this not hold for all methods i.e. less precision would act as rounding therefore less updates."
>
> We respectfully disagree with the hypothesis that sparsity is merely a universal rounding artifact. We support Gate III with both **existing evidence in the main text** and **new experiments**.
>
> **(i) Motivation already in the main text (Table 1).** If “lower precision ⇒ rounding ⇒ fewer updates” were the full story, SFT should also look sparse. However, Table 1 shows that SFT and RLVR are trained and stored in the same bf16 format, yet SFT updates are effectively dense (modifying >80% of weights), while RLVR appears highly sparse. *This asymmetry already indicates that sparsity is not just a generic consequence of bf16 rounding.*
>
> **(ii) Mechanism already explained in Sec. 2.3.** As we note in Sec. 2.3 (*Remark on precision*), standard mixed-precision training (e.g., verl) keeps optimizer states and master weights (gradient reductions/accumulation ) in fp32 [7,8]. Small gradients are not immediately “rounded away”; they accumulate in fp32. If the learning signal in a direction is strong and consistent (as in SFT), these accumulations will eventually flip the bf16 weight. *The fact that many weights never flip in RLVR therefore points to a systematic optimization bias (Gates I+II) keeping those directions small, rather than simple truncation.*
>
> **(iii) New validation: bf16 → fp16 → fp32 sweep.** To further decouple precision from the underlying bias, we run a bf16/fp16/fp32 sweep (; see Table R1 in Response to Reviewer jERv). *As precision increases, visible sparsity vanishes, but the structural learning bias remains:* even in fp32, it still shows near-identical singular values and minimal rotations, proving the updates are routed away from directions destroying the principal structure of weights.

---

> ### Author Response · Authors · 2025-11-26
> **Response to Reviewer QWqF part3**
>
> ## Q.4 Distinction from Mukherjee et al. [1]
> > “Similar insights than [1]... subnetwork discovery is a stronger claim.”
>
> We clarify our contribution in Q.0. Here, we repeat our relationship to [1] regarding four key points:
>
> **(i) Phenomenon vs. mechanism.** Mukherjee et al. [1] observe that RL updates look sparse but explicitly state they lack a theoretical explanation. We provide that missing explanation (The Three-Gate Theory).
>
> **(ii) Post-hoc vs. predictive.** [1] identifies subnetworks *post-hoc* (analyzing checkpoints after training). Our theory is predictive with a prior whihc regions is trageted or not targeted in RL learning process withour any training/assessing to gradients.
>
> **(iii) Sparsity vs. structural bias.**  Our results show that sparsity is a **read-out** of a deeper structural learning bias: RLVR preserves spectra and avoids principal weights. This geometry-aligned update pattern persists even when sparsity disappears (e.g., in fp32), indicating that the core phenomenon is the **off-principal, spectrum-preserving bias**, not sparsity itself.
>
> **(iv) Direct connection to [1]’s stated limitations.**  [1] explicitly lists “early identification of the sparse subnetwork” and “uncovering theoretical explanations for the update sparsity” as limitations and future work. Our paper advances the field precisely by addressing this open question.
>
> ## Q.5 Terminology clarification (“LLM-generated text”)
>
> > “Too much LLM generated text… use of ‘Optimization Compass’… ‘sub-/super-random’ instead of implicit regularization or correlated.”
>
> We respectfully clarify that these terms are **intentional and technically motivated**, not artifacts of LLM-generated text.
>
> **(i) “Optimization compass” vs. “implicit regularization.”**
> We introduce *“implicit RL’s compass”* in Fig. 1 and introduction as a metaphor for **directionality**: for the same pretrained model, RLVR consistently steers updates into specific regions**, regardless of dataset or RL variant. Classical “implicit regularization” typically emphasizes **magnitude shrinking** (e.g., small ‖Δθ‖). Our focus is Gate II: **where** in weight space the updates go, not just how small they are. The “compass” term is simply a concise way to summarize this directional steering bias.
>
> **(ii) “Sub-/super-random” vs. “correlated.”**
> In our overlap analyses(Sec. 4.2 defintion of Metrics.), we compare two binary masks against a **random-overlap baseline with fixed cardinality**.
> - We call a pair **“sub-random”** if its overlap is *below* the expected overlap of two random masks of the same size  $\text{Overlap} < \mathbb{E}[\text{Random overlap}]$, and **“super-random”** if it is *above* this baseline.
> - “Correlation” (Pearson/Spearman) is defined for continuous vectors and does not naturally incorporate the fixed-size constraint of our masks.
>
> ## Q.6 Comparison with Subgroup Fine-tuning [2, 3, 4]
> > Reviewer Q7: "More relevant finetuning methods... [2,3,4]. How would RL compare?"
>
> We thank the reviewer for pointing out these works. While [2,3,4] design subset-training heuristics for efficiency (mainly in SFT), our paper aims to characterize the intrinsic, emergent structure of RLVR updates from a theory perspective. In particular:
>
> **(i) Structure matters for RLVR.** Contradicting the "random subset is enough" view of SpaRTA [3], we show in Fig. 7 / Tab. 3 that the specific subset matters significantly. The principal-only masks perform significantly worse than their complementary counterparts at the same sparsity level (50%), and our selected mask also outperforms random baselines. This confirms RLVR has a strong learning bias towards a specific set of parameters.
>
> **(ii) Theory-driven prediction vs. Gradient heuristics.** Methods like MicroAdam [2] and NanoAdam [4] choose parameters based on gradient/magnitude heuristics. Our theory instead provides a training-free rule: in Sec. 4.4 we build a "safe mask" using only the pre-trained singular structure (principal vs. non-principal) . Unlike Mukherjee et al. [1] (post-hoc) or [2,4] (gradient tracking), we offer a predictive principle for which weights RLVR prefers to change.
>
> **(iii) RL works in a different optimziation regimes (off-the principal weights), so the SFT-era recipes may not align with RL dynamics (as in the SpaRTA [3])**.
>
> * **New Evidence (PiSSA):** To validate this, we added an experiment with PiSSA, a LoRA variant (referenced in the recent "Thinking Machine" blog as a promising alternative for LoRA) that specifically trains the top-$r$ principal weights. We found that PiSSA does not align with RL, pushing updates towards the undesirable region (principal weights) and leading to training instability/collapse. This confirms that recipes designed for SFT do not match the geometry-induced optimization bias of RLVR. (See details in updated manuscript Sec 5.2).

---

> > ### Author Response · Authors · 2025-11-26
> > **Reference**
> >
> > **Reference**:
> > [1] Mukherjee, Sagnik et al. “Reinforcement Learning Finetunes Small Subnetworks in Large Language Models.” ArXiv abs/2505.11711 (2025): n. pag.
> >
> > [2] Modoranu, Ionut-Vlad et al. “MicroAdam: Accurate Adaptive Optimization with Low Space Overhead and Provable Convergence.” ArXiv abs/2405.15593 (2024): n. pag.
> >
> > [3] Rios, Jesus et al. “Sparsity May Be All You Need: Sparse Random Parameter Adaptation.” ArXiv abs/2502.15975 (2025): n. pag.
> >
> > [4] Zhou, Chao et al. “Pay Attention to Small Weights.” ArXiv abs/2506.21374 (2025): n. pag.
> >
> > [5] Gunasekar, Suriya et al. “Characterizing Implicit Bias in Terms of Optimization Geometry.” ArXiv abs/1802.08246 (2018): n. pag.
> >
> > [6] Mroueh, Youssef. “Reinforcement Learning with Verifiable Rewards: GRPO's Effective Loss, Dynamics, and Success Amplification.” ArXiv abs/2503.06639 (2025): n. pag.
> >
> > [7] https://github.com/volcengine/verl/blob/8ed5bafc052bd8f172b8a4a0e777c028db1f48bd/verl/workers/fsdp_workers.py#L461-L469
> >
> > [8] https://docs.fast.ai/callback.fp16.html#the-solution-mixed-precision-training
> >
> > [9] https://thinkingmachines.ai/blog/lora/
> >
> > [10] Meng, Fanxu, Zhaohui Wang, and Muhan Zhang. "Pissa: Principal singular values and singular vectors adaptation of large language models." NeurIPS, 2024

---

### Official Review · Reviewer_6ikK · 2025-10-29

**Soundness:** 2
**Presentation:** 2
**Contribution:** 2
**Rating:** 4
**Confidence:** 3

**Summary:**

The paper attributes the sparse appearance of RL updates to three factors: (i) an on-policy KL anchor, (ii) the model’s pretrained geometry, and (iii) the use of bf16, which renders many small parameter updates numerically invisible. The analysis shows that RL maintains relatively stable spectral properties, induces less subspace rotation than SFT, and tends to avoid modifying principal weights—whereas SFT more frequently affects principal directions in the parameter space.

**Strengths:**

- Clear narrative supported by strong evidence.
The paper presents a coherent story backed by quantitative evidence. The results show strong cross-run overlap (Jaccard/consensus metrics), indicating structured consistency rather than random speckle patterns.

- Insightful PEFT takeaway.
The analysis demonstrates that “principal-only” masks underperform, while non-principal or low-magnitude masks better align with dense KL trajectories and maintain accuracy closer to the dense baseline.

- Precision as an interpretive lens.
The work highlights how numerical precision affects interpretability: with fp32 storage or larger learning rates, the observed “sparsity” largely disappears. This suggests that bf16 precision mainly influences the visibility rather than the existence of updates.

- Cross-family generalization.
Although most evidence is centered on Qwen, the paper includes additional results on LLaMA and Mistral, indicating that the observed phenomena are not model-specific.

**Weaknesses:**

- Missing precision sweep.
The paper does not include results for fp16 or bf8. A small precision ablation would help validate the claim that the observed sparsity is primarily an artifact of numerical representation.

- Incomplete SFT comparison.
The authors claim that SFT targets principal weights, yet they do not present SFT update-mask overlap plots at the same granularity as those for RL. Including these would substantiate the contrast more rigorously.

- Limited cross-model evaluation.
While the paper includes some non-Qwen results, these experiments are lighter and less comprehensive. More like-for-like evaluations on LLaMA and Mistral would strengthen claims of generality.

- Reproducibility concerns.
The analysis code—particularly for computing masks, rotations, and subspace probes—does not appear to be released. Open-sourcing these components would enhance reproducibility and transparency.

**Questions:**

- If SFT is trained with a KL-to-base regularization term, does its update-mask overlap with principal weights decrease toward RL’s pattern? Could the authors include SFT-mask overlap visualizations for direct comparison?

- Could the authors include fp16 and bf8 storage or accumulation settings to further evaluate the “precision as lens” hypothesis?

- Could the authors share code or detailed settings for the right-multiplication block-orthogonal rotations (e.g., per-head grouping, GQA configurations), as well as your exact procedure for computing subspace angles? Clarifying these would help reconcile potential discrepancies in replication.

---

> ### Author Response · Authors · 2025-11-26
> **Response to Reviewer 6ikK part1**
>
> # Response to Reviewer 6ikK
>
> We thank the reviewer for their detailed assessment and for recognizing the *clear narrative*, *strong evidence*, and *insightful PEFT takeaway*. We appreciate the constructive suggestions and respond point-by-point below.
>
>
> ## Q.1 Precision sweep & “precision as lens” hypothesis
> > "Include results for fp16 or bf8... to validate the claim that sparsity is primarily an artifact."
>
> Great suggestion! We agree that a precision sweep strengthens the “precision as lens” claim. In the revision, we add:
>
>
> **(i) New Evidence: Precision Sweep (BF16 $\to$ FP16 $\to$ FP32).** As verl doesn't support bf8 yet, we train RLVR runs in bf16, fp16, and fp32 on DS-Qwen-1.5B and LLaMA-3.2-3B and measure both update sparsity and geometry metrics (singular-value drift and principal-subspace rotation). As shown in Table R4 below, whole-model and one representative layer (Q_proj)'s sparsity collapse with higher precision (bf16→fp16→fp32), while Q_proj principal angles and singular-value drift remain on the order of 10⁻¹ degrees and 10⁻⁴, supporting our “precision is a lens, not the cause” claim.
>
>
> **Table R4 (summary). Precision ablation on DS-Qwen-1.5B and LLaMA-3.2-3B (Q_proj).** (More per-layer results are in Table R1 in the response to Reviewer jERv.)
>
> | Precision | DS-Qwen-1.5B | | |  | LLaMA-3.2-3B                 |  |    | |
> |----------:|--------------------------------:|----------------------:|-----------------------:|-----------------------:|---------------------------------:|----------------------:|-----------------------:|-----------------------:|
> |           | Sparsity (%, whole)            | Q_proj sparsity (%)   | Q_proj θ_max (°)       | Q_proj Δσ_max          | Sparsity (%, whole)             | Q_proj sparsity (%)   | Q_proj θ_max (°)       | Q_proj Δσ_max          |
> | bf16      | 82.38 | 79.78| 0.47| 8.9e-5                 | 68.76                           | 68.20                 | 0.78                   | 1.99e-4                |
> | fp16      | 35.16  | 26.89    | 0.79 | 1.65e-4                | 18.65                           | 16.02                 | 1.01                   | 2.29e-4                |
> | fp32      |  3.25                           |  0.00                 | 0.62                   | 1.54e-4                |  0.00                           |  0.00                 | 0.74                   | 3.29e-4                |
>
> **(ii) Mechanism (already in Sec. 2.3).**  As noted in Sec. 2.3, the **mixed precision training**(as in verl) maintains optimizer states and master weights (gradient reduction/accumulation) are fp32. Small updates are accumulated in fp32, not immediately rounded away. If the learning signal were strong and persistent (as in SFT), these accumulations would eventually flip bf16 weights. The fact that many weights never flip in RLVR, despite fp32 accumulation, points to a **systematic optimization bias (Gates I+II)** rather than “less precision ⇒ fewer updates.”
>
> **(iii)Conclusion:** Together, the bf16/fp16/fp32 sweep and the verl implementation details support Gate III: precision mainly acts as a **lens** that makes the KL+geometry bias visible as sparsity, rather than causing it.
>
> ## Q.2 Incomplete SFT Comparison
> > "SFT update-mask overlap plots"
>
> We respectfully clarify that **a direct "mask overlap" comparison is not applicable because SFT updates are dense, effectively changing almost all parameters** There is no "sparse SFT mask" to visualize.
>
> Instead, we compare SFT and RLVR via **where SFT effectively acts**, using principal weights and spectra:
>
> **(i) Principal Weights as the SFT Target Proxy**
> Since SFT moves nearly all weights, we define its “targeted region” using the **principal weights** \(M_{\text{princ}}\) from recent PEFT work [2]: the top-\(p\%\) largest-magnitude weights inside the top-\(r\) singular subspace.
> * Why this proxy works: Literature [2] identifies these as the "load-bearing" weights that drive effective SFT when has limited parameter bugdet, i.e., sparse fine-tuning. Perturbing them causes significant performance drops (across perplexity, next-token prediction, and reasoning in their Fig.2), confirming they lie on high-curvature directions of the optimization landscape.
> * Validation via PiSSA: PiSSA [3] further supports this by **explicitly initializing LoRA adapters on these principal components (top-r sinular subspace)** to accelerate SFT, indicating this is the region SFT relies on most.
>
> **(ii) Spectral Destruction (SFT) vs. Preservation(RL)** This contrast is empirically confirmed by our Spectral Checks (Fig 4). SFT causes significant singular value drift and subspace rotation, proving it disrupts (and therefore targets) the principal components. RLVR, in contrast, preserves this structure.
>
> Taken together, the principal-weight proxy + spectral analysis provide a more meaningful comparison than a binary SFT mask: **SFT targets and rotates principal directions**, whereas **RLVR largely avoids them and preserves geometry**.

---

> ### Author Response · Authors · 2025-11-26
> **Response to Reviewer 6ikK part2**
>
> ## Q3. SFT with KL-to-base vs. intrinsic RL bias
> > " Does SFT mask overlap with principal weights with KL-to-base regularization term?"
>
> We address this with two points:
>
> **(i) Standard SFT does not use a KL loss.**
> Standard SFT objectives aim to fit an external data distribution; they do not typically include a KL-to-base penalty.
>
> **(ii) The Bias is Intrinsic to RL Dynamics, Not Explicit KL Loss.**
> We emphasize that RL’s off-principal behavior is not caused by adding an explicit KL loss. In fact, our main experiments use **DAPO**, which has **no explicit KL term** in the loss. Yet DAPO still exhibits the same “implicit compass” behavior (KL-proximal policy moves, off-principal parameter updates, preserved spectra). This matches Gate I: on-policy RL (PPO/GRPO/DAPO-style) inherently yield **KL-proximal update step**, regardless of whether the KL appears explicitly in the loss.
>
>
> ## Q4. Cross-model generalization (LLaMA / Mistral)
>
> > “More like-for-like evaluations on LLaMA and Mistral.”
>
> We respectfully note that a perfect "cross-task" sweep (as done for Qwen in our paper) is challenging: their **base checkpoints are relatively weak at reasoning** and typically require mid-training[4], which introduces confounders when comparing “before/after RL” in a controlled way. So we provide some but not many non-Qwen results.
>
> Within these constraints, we strengthen our evidence in two ways:
>
> **(i) Same model, different RL algorithms (consensus check on LLaMA-3.2-3B).**
> We compare RL variants on **LLaMA-3.2-3B** on MATH datasets and apply our update-mask and spectral diagnostics. As shown in Fig. 9 (included in the appendix of revised version), we observe **consistent, structured update patterns across algorithms**: the same regions are repeatedly selected, similar to what we observe on DS-Qwen-1.5B. This cross-run consensus further proves that the “implicit compass” is not an artifact of a particular architecture.
>
> **(ii) Inclusion in new mechanism experiments.**
> We also include **LLaMA-3.2-3B** in our new **precision sweep (bf16/fp16/fp32)** and **LoRA/PiSSA experiments**. In all cases, LLaMA shows the same structural learning bias as Qwen: visual sparsity depends on precision, but the core pattern of spectral preservation and off-principal updates remains invariant.
>
> Together, these results support that the structural learning bias we describe is a **property of RL optimization on pretrained LMs**, rather than a Qwen-specific phenomenon.
>
>
>
> ## Q.5 Reproducibility and Code Release
> > "Share code... for right-multiplication block-orthogonal rotations."
>
> We are fully committed to reproducibility. We will release the complete codebase upon acceptance.
>
> **(i) Immediate Verification:** To facilitate your assessment during the review process, we have provided a minimal functional script (see updated supplementary) that allows you to:
> * Compare weight matrices between the RL run pair (e.g., NVIDIA-ProRL vs. DeepSeek-R1-Distill-Qwen-1.5B) and the SFT run pair (e.g., DeepSeek-R1-Distill-Qwen-1.5B vs. Qwen-1.5B)
> * Reproduce the bf16-aware sparsity calculation and Principal Angle rotation metrics reported in the paper.
>
> **(ii)Clarification on Rotation Method:** We respectfully note that the mathematical procedure for the orthogonal rotations is already detailed in **Appendix C.3.** Our implementation is adopted directly from the public SpinQuant codebase [1]. We use their standard block-orthogonal matrices (e.g., Hadamard) to "scramble" the weight geometry while preserving the mathematical equivalence of the linear layer's output.
>
> ------
> **References:**
>
> [1] SpinQuant: https://github.com/facebookresearch/SpinQuant
>
> [2] Liu, Zihang, et al. "LIFT the Veil for the Truth: Principal Weights Emerge after Rank Reduction for Reasoning-Focused Supervised Fine-Tuning." ICML 2025
>
> [3] Meng, Fanxu, Zhaohui Wang, and Muhan Zhang. "Pissa: Principal singular values and singular vectors adaptation of large language models." NeurIPS 2024
>
> [4] Wang, Zengzhi, et al. "Octothinker: Mid-training incentivizes reinforcement learning scaling." arXiv preprint arXiv:2506.20512

---

### Official Review · Reviewer_BtSB · 2025-11-01

**Soundness:** 3
**Presentation:** 4
**Contribution:** 2
**Rating:** 4
**Confidence:** 3

**Summary:**

This paper investigates why Reinforcement Learning with Verifiable Rewards (RLVR), achieves major reasoning improvements in large language models while modifying only a small portion of parameters. The authors propose that this apparent sparsity reflects an implicit optimization bias—an “implicit compass” guiding updates.

They introduce a Three-Gate Theory:
- Anchoring (Gate I): KL regularization constrains updates near the base policy.
- Geometry (Gate II): The pretrained model’s structure steers updates toward low-curvature regions, avoiding principal weights.
- Precision (Gate III): bfloat16 precision amplifies apparent sparsity by hiding small updates.

Extensive experiments across models and RL variants show that RL preserves spectral structure, avoids principal weights, and exhibits stable, geometry-aligned update patterns. Causal interventions confirm that model geometry drives this bias. The work reframes sparsity as a byproduct of structured optimization.

**Strengths:**

1. Fresh perspective on RL updates:

The paper gives a genuinely new way to think about why RL updates look sparse. The “implicit compass” idea connecting KL constraints, model geometry, and precision is original and thought-provoking.

2. Strong empirical evidence:

The experiments are thorough and consistent across models. I especially liked the causal intervention tests — they make a convincing case that the observed patterns aren’t just artifacts.

**Weaknesses:**

1. General claim but limited evaluation:

The paper makes a rather broad theoretical claim about the nature of RL updates under KL constraints, arguing that the observed sparsity pattern and directional bias are universal properties of such optimization dynamics. However, the experimental validation is restricted to RLVR applied to large language models. This raises questions about how general the findings really are.

To strengthen the claim, it would be valuable to test whether the same “implicit compass” behavior appears in other RL formulations or domains. For instance, examining smaller-scale robotic control tasks, or an imitation learning setup where a pretrained policy is fine-tuned either through RL or supervised learning, could reveal whether this bias is a general feature of KL-regularized RL or specific to language models.

2. Unclear practical takeaway:

While the proposed “implicit compass” theory is conceptually appealing and provides a new lens to interpret the geometry of RL updates, its practical implications remain underexplored. The paper convincingly shows that RL updates behave differently from supervised fine-tuning, but it does not clearly explain how this understanding can be used to improve algorithm design.

For example, could this theory inform new geometry-aware optimization schemes, selective parameter updates, or adaptive KL regularization strategies? Are there concrete steps to make RL training more stable or sample-efficient based on this insight? Clarifying how the theory translates into tangible algorithmic guidance, or even including a small proof-of-concept experiment would significantly enhance the practical impact of the paper.

**Questions:**

1. Relation to recent sparsity work:

Recent papers in the RL literature [1,2] have shown that introducing explicit sparsity into model weights can improve scaling performance. According to your Three-Gate Theory, sparsity already emerges naturally as a byproduct of the RL update process constrained by KL regularization. How should we reconcile these findings? Why does imposing stronger, explicit sparsity still seem to help, if sparsity is already an implicit consequence of RL optimization? Does your theory suggest that RL inherently requires stricter or more structured forms of sparsity beyond what naturally emerges?

[1] Network Sparsity Unlocks the Scaling Potential of Deep Reinforcement Learning, ICML 2025 (Oral)

[2] Rethinking the Role of Dynamic Sparse Training for Scalable Deep Reinforcement Learning, arXiv 2025


I lean toward a weak reject at this stage. The paper presents an interesting and original theoretical perspective on why RL updates appear sparse, but the practical significance are not yet convincing enough. That said, I am open to improving my score if the authors can clearly demonstrate why this theory matters; for example, how the proposed “implicit compass” insight could meaningfully guide the design of more efficient or better RL algorithms.

---

> ### Author Response · Authors · 2025-11-26
> **Response to Reviewer BtSB part1**
>
> # Response to Reviewer BtSB
>
> We sincerely thank Reviewer BtSB for the insightful feedback and for recognizing our "Implicit Compass" perspective as "original and thought-provoking" with strong empirical backing. We appreciate the opportunity to clarify the scope of our claims and demonstrate the concrete practical utility of our theory.
>
> ----
>
> In response, we have extensively revised the manuscript (see updated PDF). We added expanded evaluations on Agents, RLHF, and Robotics (Section 4.4) and a dedicated section on "Theory-Guided Learning Algorithms" (Section 5). This section details how our geometric insights directly inform algorithm design, deriving Safe Masks and correctly predicting the failure of SFT heuristics (e.g., PiSSA, a 'promising' next step suggested in the recent Thinking Machines LoRA blog[2].) in RL.
>
> ----
>
> Please check our detailed response here.
>
>
> ## Q.1  Extended analysis on more tasks (Agents, RLHF, Robotics)
>
> > "Test whether the same 'implicit compass' behavior appears in... robotic control tasks... to reveal whether this bias is a general feature."
>
> We appreciate this suggestion.
> We first clarify our **core theoretical claim**: the Structural Learning Bias (updates steered toward off-principal subspaces with preserved spectra) is not specific to language foundation models. Rather, it is a consequence of applying KL-constrained updates (Gate I) to pre-trained Foundation Models that possess a stable, well-initialized optimization landscape (Gate-II).
>
> To validate this, we extended our diagnostics to three distinct regimes beyond standard math/code tasks (detailed in revised Sec. 4.4):
> * Multi-turn Agents: Tool-use tasks via AgentFlow and SkyRL.
> * RLHF: DPO and SimPO on Llama-3 (Instruction Following).
> * Embodied AI (Robotics): Embodied-r1[6] on general robotic manipulation and SimpleVLA-RL [7] on LIBERO tasks [8].
>
> **Case Study: Robotics (SimpleVLA-RL)**. We performed a deep dive into the SimpleVLA-RL model, which fine-tunes an OpenVLA backbone (combining ViT and LLM) on manipulation tasks. We compared a standard SFT checkpoint (Haozhan72/Openvla-oft-SFT-libero10-trajall) against an RL-finetuned checkpoint (Haozhan72/openvla-oft-libero10-traj1-rl) (perform RL with a warm-started SFT).
>
> **Conclusion**: The results in Table R3 confirm our theory holds for robotics. The RL-tuned model maintains minimal rotation and stable singular values, confirming that KL-constrained RL preserves the pre-trained spectral structure by orthogonalizing updates against principal components. SFT, conversely, destroys this structure.
>
> **Table R3: Robotics Case Study.** Comparison of SFT vs. RL on the same backbone. RL preserves the spectral structure (low $\theta$, minimal $\Delta \sigma$) across both modalities, while SFT destroys it. We consistently use the 13-th block from both the language and the vision parts, and report partial layers due to space limit. We show the full rotation range for for both left ($U$) and right ($V$) spaces.
>
> |  | Module | Metric | SFT | RL |
> |---|---|---|---|---|
> | Whole model |  | Sparsity (%) | 3.46% | 35.04% |
> | Language part | Q proj | Sparsity (%) | 0.96% | 35.35% |
> |  |  | $\theta$ Rot. (°) | (5.0057-88.4482) / (4.9644-89.2156) | 0.2219-4.0525 / (0.2118-4.0449) |
> |  |  | Max $\sigma$ Diff | 2.7732 | 0.0009 |
> |  | O proj | Sparsity (%) | 0.83% | 27.77% |
> |  |  | $\theta$ Rot range (°) | (6.3856-89.8569) / (6.3523-89.5472) | (0.4296-22.1034)/(0.4170-22.1084) |
> |  |  | Max $\sigma$ Diff | 2.063 | 0.0006 |
> |  | up_proj | Sparsity (%) | 0.81% | 30.80% |
> |  |  | $\theta$ Rot. (°) | (6.8117-89.8272) / (6.2930-89.7190) | (0.4457-42.3534)/(0.3807-42.3525) |
> |  |  | Max $\sigma$ Diff | 4.269 | 0.0008 |
> | Vision part: Featurizer (a ViT model) | qkv | Sparsity (%) | 0.92% | 39.57% |
> |  |  | $\theta$ Rot. (°) | (4.1584-82.8790) /(3.2640 - 83.2722) | (0.2167-3.1509) / (0.1323 - 3.1504) |
> |  |  | Max $\sigma$ Diff | 0.3454 | 0.0009 |
> |  | fc1 | Sparsity (%) | 0.85% | 37.91% |
> |  |  | $\theta$ Rot. (°) | (4.4920-89.6873) / (3.3385-89.6818) | (0.2471-3.2063) / (0.1391-3.1972) |
> |  |  | Max $\sigma$ Diff | 1.355 | 0.0009 |
> | Vision part:  fused_featurizer (a ViT model) | qkv | Sparsity (%) | 0.81% | 30.45% |
> |  |  | $\theta$ Rot. (°) | (5.1612-88.4576) / (4.1934 - 85.9854) | (0.3212-4.6259) / (0.2121-4.6205) |
> |  |  | Max $\sigma$ Diff | 0.7716 | 0.0008 |
> |  | fc2 | Sparsity (%) | 0.72% | 32.07% |
> |  |  | $\theta$ Rot. (°) | (4.3909 - 87.8277) / (5.4537 - 89.3361) | (0.2307-6.4751) / (0.3290 - 6.4797) |
> |  |  | Max $\sigma$ Diff | 0.7809 | 0.0026 |

---

> ### Author Response · Authors · 2025-11-26
> **Response to Reviewer BtSB part2**
>
> ## Q.2 Relation to Sparsity in Deep RL [3, 4]
>
> > "How should we reconcile these findings [with explicit sparsity in Deep RL]? ... Does your theory suggest that RL inherently requires stricter or more structured forms of sparsity?"
>
> We appreciate these references. We clarify that "sparsity" plays fundamentally different roles in traditional Deep RL (DRL) compared to RL fine-tuning of Foundation Models (FM).
>
> **(i) DRL Sparsity (for Plasticity):** In the DRL literature [3, 4, 5], explicitly imposed sparsity is an intervention designed to solve specific optimziation pathologies, such as capacity collapse and plasticity loss, during scaling DRL (which is small with ~1M to 362M [3, 4, 5]. Here, sparsity serves as an inductive bias to prevent the model from "forgetting how to learn," thereby enabling parameter scaling.
>
> **(ii) FM Sparsity (Emergent Artifact of Geometry):**: In contrast, Foundation Models possess robust, pre-trained representations (> billions of parameters). There is no clue showing that it has the plasticity loss observed in DRL.
>
> * **Cause of Sparsity**: The sparsity we observe is not an engineered fix, but a readout of the model's structural learning bias, which steers updates away from high-curvature and principal weights, due to limited precision. With precision increase (see Q.1 in rebuttal to Reviewer jERv), sparsity diminishes.
> * **Real geometry constraint to honor** Consequently, our theory does not suggest imposing explicit or stricter sparsity (as in DRL). Instead, it prescribes geometric alignment: effective RL algorithms for FMs must respect the optimization landscape by avoiding principal weights.
>
> We will include the discussion of sparsity between DRL and RL for foundation model fine tuning in the future version.

---

> ### Author Response · Authors · 2025-11-26
> **Response to Reviewer BtSB part3**
>
> ## Q.3 Practical Takeaway: From White-Box Understanding to Tangible Algorithm Design Guidance
>
> > "Clarifying how the theory translates into tangible algorithmic guidance... would significantly enhance the practical impact."
>
> We agree that a good theory must not only explain observations but also inform design. Our work offers two layers of practical value:
>
> **(i) Towards a white-box understanding of RL dynamics**
> *We provide the first parameter-level characterization of how parameters evolve when RL finetunes foundaiton models.*  Our analysis reveals that this process is not random, but is governed by **a persistent structural optimization bias**: *for a fixed pre-trained model, RL updates systematically avoid principal weights, localizing instead to off-principal, spectrum-preserving subspaces*.
>
> This mechanistic offers a fresh explanation for distinct phenomena such as, why RL updates appear sparse and why RL forgets less.
>
>
>
> **(ii) Prescriptive guidance for algorithm design**
>
>
> Crucially, we show that **RL operates in a different optimization regime from SFT**: RLVR learns in off-principal directions while preserving layer spectra, whereas SFT aggressively rotates and modifies principal components.
>
> > This unique learning dynamics implies that effective RL learning algorithms must be **geometry-aligned**, honoring the off-principal bias of the RL landscape. **SFT-era PEFT heuristics, tuned to SFT dynamics(principal-direction updates), could be flawed.**
>
>
> We validate this tangible guideline with two examples:
>
>
> *  **(a)The "Safe Mask" (RL-Specific sparse fine-tuning).**
> Our theory allows us to construct a "Safe Mask" ($M_{\text{safe}} = M_{\text{low}} \cup M_{\text{princ}}^{c}$) **using only pre-trained geometry—zero training required.** Training on this mask matches dense RLVR performance, whereas SFT-style principal masks degrade performance.
>
>
>
> * **(b)Geometry-Alignment (Refuting SFT Heuristics)**
> A recent Thinking Machines blog highlights the success of LoRA [2] for RL and suggests **PiSSA** as a promising next step, based on its strong performance in SFT.
>
>   - **PiSSA background:** unlike standard LoRA, PiSSA **initializes adapters from the top-r principal components** of the pretrained weights, explicitly targeting principal directions.
>   - **Our prediction:** since RLVR **avoids principal weights**, our theory predicts that PiSSA is **geometrically misaligned** for RL, because it forces updates into exactly those principal parts RLVR is trying to preserve.
>
>   In Sec. 5.2 (Figs. 19–20), we test this prediction on **DS-Qwen-1.5B** and **LLaMA-3.2-3B** under the Thinking Machines settings. As predicted:
>   - PiSSA shows **no clear gain** over LoRA in RL, and **PiSSA is noticeably more unstable and prone to early collapse**, while standard LoRA remains stable.
>
>
> **Conclusion Together,** our work charts a path toward a white-box understanding of RLVR and the design of geometry-aware, RLVR-native learning algorithms, rather than repurposed SFT-era heuristics.
>
> ------
>
> **Reference**:
> [1] Meng, Fanxu, Zhaohui Wang, and Muhan Zhang. "Pissa: Principal singular values and singular vectors adaptation of large language models." NeurIPS 2024
>
> [2] https://thinkingmachines.ai/blog/lora/
>
> [3] Network Sparsity Unlocks the Scaling Potential of Deep Reinforcement Learning, ICML 2025 (Oral)
>
> [4] Rethinking the Role of Dynamic Sparse Training for Scalable Deep Reinforcement Learning, arXiv 2025
>
> [5] Lee, Hojoon, et al. "Simba: Simplicity bias for scaling up parameters in deep reinforcement learning." arXiv preprint arXiv:2410.09754 (2024).
>
> [6] Yuan, Yifu, et al. "Embodied-r1: Reinforced embodied reasoning for general robotic manipulation." arXiv preprint arXiv:2508.13998 (2025).
>
> [7] Li, Haozhan, et al. "Simplevla-rl: Scaling vla training via reinforcement learning." arXiv preprint arXiv:2509.09674 (2025).
>
> [8] Liu, Bo, et al. "Libero: Benchmarking knowledge transfer for lifelong robot learning." Advances in Neural Information Processing Systems 36 (2023): 44776-44791.

---

### Official Review · Reviewer_jERv · 2025-11-04

**Soundness:** 3
**Presentation:** 4
**Contribution:** 4
**Rating:** 8
**Confidence:** 3

**Summary:**

The paper asks why RLVR seems to improve reasoning models while touching surprisingly few parameters. The authors argue that observed “sparsity” is mostly an artifact of an implicit optimization bias (“implicit compass”) rather than true selectivity. Their Three-Gate Theory combines: (1) an on-policy KL leash (Anchor Gate), (2) pretrained geometry with easy vs. principal directions (Geometry Gate), and (3) bf16 precision masking tiny updates (Precision Gate). They show RLVR preserves spectral structure and avoids principal weights compared to SFT, and a causal intervention that scrambles layer geometry disrupts the effect. Finally, masks derived from this perspective approach dense training, pointing toward RL-specific PEFT.

**Strengths:**

- Mechanistic clarity: The Three-Gate framing connects training constraints to parameter movement in a way practitioners can reason about.
- Causal test: Scrambling/rotating geometry to break the bias goes beyond correlation.
- Better measurement: The bf16-aware sparsity metric corrects prior over-interpretations.
- Practical value: The mask results suggest RL-specific PEFT is both necessary and feasible.

**Weaknesses:**

- Curvature proxy: “Principal weights” stand in for high curvature. Even a coarse Fisher-diagonal estimate would help validate this mapping.
- Precision disentanglement: It’s not fully shown whether bf16 is merely a lens vs. a contributing cause. A strict fp32-only run (weights + optimizer + no casts) would clarify Gate III.
- Generalization breadth: Deep analysis centers on Qwen/DeepSeek. Quantitative replication on Llama/Mistral would support universality claims.

**Questions:**

- Beyond bf16: If you train and store everything in fp32, do the consensus stripes and spectral preservation persist?
- Curvature validation: What is the correlation between your principal-weight masks and per-parameter Fisher diagonals (even on a small batch)?
- Gate coupling: How do the stripes change as you sweep KL β (explicit) or clipping ε (implicit leash)? A monotone response would directly tie Gate I to Gate II.

---

> ### Author Response · Authors · 2025-11-26
> **Response to Reviewer jERv part1**
>
> We thank the reviewer for the encouraging assessment and for highlighting the "mechanistic clarity" and "practical value" of our Three-Gate Theory. Below we address the raised concerns.
>
> ## Q.1 Precision is a lens, not the cause.
> > "It’s not fully shown whether bf16 is merely a lens... A strict fp32-only run (weights + optimizer + no casts) would clarify Gate III."
>
> We followed this excellent suggestion and added RLVR runs with FP32 and FP16 training [1] on DS-Qwen-1.5B (DeepMath-103K [3]) and Llama-3.2-3B (MATH [4]), using the original BF16 setup as a baseline.
>
> **(i) Precision Sweep (BF16 $\to$ FP16 $\to$ FP32).**
> The results (Table R1) confirm the "Lens" Hypothesis:Sparsity (The Symptom): As precision increases, visual sparsity drops dramatically. For the whole model, sparsity vanishes from ~82% (BF16) to 0.00% (FP32). This confirms that BF16 acts as a filter that "hides" small-magnitude updates, while FP32 captures them.
> * **Geometry (The Mechanism): **Crucially, the Structural Learning Bias remains invariant. Even in the dense FP32 updates, the Max Singular Value Difference ($\Delta \sigma$) remains remarkably stable at the $10^{-4}$ magnitude, and subspace rotation ($\theta$) changes minimally.
>
> **(ii) Conclusion.** This definitively proves Gate III: precision acts as a lens that makes the existing KL+geometry bias appear as sparsity, rather than being its root cause. The model continues to steer updates off-principal (preserving the core spectrum) regardless of whether those updates are visible (FP32) or filtered (BF16).
>
>
> **(iii)More explantion: Why precision has little impact (Sec. 2.3).**
> As noted in Sec. 2.3, in verl’s implementation, optimizer states and master weights (gradient accumulation and reduction) are stored in fp32; small gradients accumulate in fp32 and are not immediately rounded away. Thus, bf16 is not the key issue to drive the unique learning dynamics.
>
>
> Table R1: Precision Ablation Study. Comparing Sparsity vs. Geometric Stability across precisions. Note that while Sparsity collapses to near-zero in FP32, the geometric metrics ($\theta$ and $\Delta \sigma$) remain stable. We report rotation for both left ($U$) and right ($V$) spaces.
>
> |  |  | DS-Qwen-1.5B |  |  | Llama-3.2-3B |  |  |
> |---|---|---|---|---|---|---|---|
> | Module | Metric | bf16 | fp16 | fp32 | bf16 | fp16 | fp32 |
> | Whole model | Sparsity (%) | 82.38% | 35.16% | 3.25% | 68.76% | 18.65% | 0.00% |
> | Q proj | Sparsity (%) | 79.78% | 26.89% | 0.00% | 68.20% | 16.02% | 0.00% |
> |  | Max $\theta$ Rot. (°) | 0.4731/0.4735 | 0.7871/0.7881 | 0.6215/0.6209 | 0.7838/0.7826 | 1.0099/1.0107 | 0.7426/0.7431 |
> |  | Max $\sigma$ Diff | 8.90E-05 | 1.65E-04 | 1.54E-04 | 1.99E-04 | 2.29E-04 | 3.29E-04 |
> | K proj | Sparsity (%) | 77.40% | 26.17% | 0.00% | 74.10% | 20.40% | 0.00% |
> |  | Max $\theta$ Rot. (°) | 0.0685/0.3626 | 0.0626/0.4049 | 0.0685/0.4182 | 2.0905/2.0921 | 2.2828/2.2847 | 1.9078/1.9082 |
> |  | Max $\sigma$ Diff | 7.90E-05 | 1.27E-04 | 1.44E-04 | 2.97E-04 | 2.99E-04 | 1.66E-04 |
> | V proj | Sparsity (%) | 76.89% | 24.67% | 0.00% | 51.32% | 8.82% | 0.00% |
> |  | Max $\theta$ Rot. (°) | 0.0593/0.2846 | 0.0626/0.3263 | 0.0626/0.4073 | 1.9608/1.9654 | 1.3452/1.3430 | 1.9078/1.9141 |
> |  | Max $\sigma$ Diff | 8.70E-05 | 1.16E-04 | 1.48E-04 | 1.26E-04 | 1.39E-04 | 1.40E-04 |
> | O proj | Sparsity (%) | 83.60% | 32.54% | 0.00% | 56.17% | 10.20% | 0.00% |
> |  | Max $\theta$ Rot. (°) | 0.3375/0.3386 | 1.3442/1.3467 | 0.5279/0.5403 | 3.2303/3.2325 | 2.9191/2.9209 | 1.5612/1.5706 |
> |  | Max $\sigma$ Diff | 6.40E-05 | 9.60E-05 | 1.16E-04 | 1.00E-04 | 8.30E-05 | 9.50E-05 |
> | MLP Up_Proj | Sparsity (%) | 80.63% | 27.00% | 0.00% | 62.98% | 12.65% | 0.00% |
> |  | Max $\theta$ Rot. (°) | 0.8339/0.8330 | 0.7836/0.7833 | 1.2045/1.2037 | 3.4001/3.3988 | 3.3315/3.3315 | 1.5000/1.5012 |
> |  | Max $\sigma$ Diff | 9.70E-05 | 1.50E-04 | 2.26E-04 | 1.23E-04 | 1.48E-04 | 2.69E-04 |
> | MLP Down_proj | Sparsity (%) | 79.85% | 26.17% | 0.00% | 62.18% | 12.36% | 0.00% |
> |  | Max $\theta$ Rot. (°) | 0.6543/0.6555 | 1.4175/1.4180 | 1.5846/1.5847 | 15.5249/15.5248 | 5.4301/5.4302 | 2.0066/2.0066 |
> |  | Max $\sigma$ Diff | 1.37E-04 | 1.50E-04 | 2.27E-04 | 2.13E-04 | 2.43E-04 | 3.26E-04 |
> | MLP Gate_proj | Sparsity (%) | 81.61% | 28.05% | 0.00% | 63.63% | 13.01% | 0.00% |
> |  | Max $\theta$ Rot. (°) | 0.3690/0.3696 | 0.6299/0.6284 | 0.5795/0.5774 | 2.6317/2.6305 | 1.6107/1.6084 | 1.3662/1.3640 |
> |  | Max $\sigma$ Diff | 9.10E-05 | 1.63E-04 | 1.68E-04 | 1.72E-04 | 2.02E-04 | 2.13E-04 |

---

> ### Author Response · Authors · 2025-11-26
> **Response to Reviewer jERv part2**
>
> ## Q.2 Justification of Principal Weights as a High-Curvature Proxy
>
> > "Correlation between principal-weight masks and per-parameter Fisher diagonals?"
>
> We appreciate this insightful question. We would like to justify our choice of Principal Weights[2] as a proxy of high-curvature directions here.
>
> **(i) Theoretical connection.**"High curvature" corresponds to large eigenvalues of the Hessian (or Fisher Information) $H$. Mathematically, a perturbation $\delta$ in such a direction induces a significant degradation in loss: $\Delta \mathcal{L} \approx \frac{1}{2}\delta^T H \delta$.
> * **Perturbation Analysis to show it is a good proxy [2]:** Liu et al. [2] performed a direct perturbation analysis, demonstrating that adding random noise to $M_{prin}$ (weights in the top-$r$ singular subspace) causes significantly larger degradation in downstream tasks (perplexity, next-token prediction, reasoning accuracy) than adding the same noise to random or magnitude-based directions. This confirms that $M_{prin}$ is highly correlated with the performance-sensitive, high-curvature parameters of the loss landscape.
>
>
> This confirms that Principal Weights[2] are indeed correlated well with performance-sensitive parameters, effectively serving as a proxy for high-curvature directions without the prohibitive cost of computing the full Hessian. This is further corroborated by the state-of-the-art PEFT method PiSSA [5], which initializes adapters on these principal components to accelerate SFT convergence. We leave the costly tracking of the second-order Fisher matrix for future experiments.
>
>
> ## Q.3 Gate coupling via KL β / clipping ε sweeps.
>
> > "How do the stripes change as you sweep KL $\beta$ or clipping $\epsilon$? A monotone response would directly tie Gate I to Gate II."
>
> Great suggestion. To study this coupling, we sweep **β** and **ε** on **LLaMA-3.2-3B (Math)**:
>
> - KL penalty:  β ∈ {0, 5e-3, 5e-2}
> - Clip-high threshold:  ε ∈ {0.28, 0.18, 0.08}
>
> Our main experiments already use **DAPO (β = 0, ε = 0.28)**, i.e., a “light leash” with no explicit KL term.
>
>
> **(i) Monotone effect of β (explicit KL).**
> We observe a strict monotonic effect in the transformer blocks. As $\beta$ increases (tightening the "leash"):
> * **Sparsity Increases:** Visual sparsity rises (e.g., Q-Proj from 68.2% to 70.4%).
> * **Rotation Decreases:** Subspace rotation drops (e.g., Q-Proj from 0.78° to 0.46°).
>
> This gives a clear monotone response: stronger KL anchoring (Gate I) makes updates **smaller and more geometry-preserving**, reinforcing Gate II.
>
>
> **(ii) A weak effect of Clipping ($\epsilon$)**:
> Sweeping ε has negligible effect in our setting: the fraction of actually clipped tokens at ε = 0.28 is already very small (~0.5%), so tightening ε further adds almost no active constraint.
>
>
> Table R2: Sensitivity Analysis. Geometric metrics on LLaMA-3.2-3B (Layer 13) across KL penalty ($\beta$) and clipping ($\epsilon$) sweeps. We report rotation for both left ($U$) and right ($V$) spaces.
> Note: A strong KL penalty ($\beta=5\text{e-}2$) tends to over-regularize the embedding and head layers, causing a regression in whole-model sparsity. We therefore report sparsity both with and without these layers to isolate the monotonic behavior of the transformer blocks.
>
> | Module | Metric | β=0 (Base) | β=5e-3 | β=5e-2 | ϵ=0.28 (Base) | ϵ=0.18 | ϵ=0.08 |
> |---|---|---|---|---|---|---|---|
> | All Modules  (w/o embedding and head) | Sparsity (%) | 64.56% | 65.90% | 67.50% | 64.56% | 63.44% | 63.39% |
> | All Modules | Sparsity (%) | 68.76% | 69.58% | 66.88% | 68.76% | 67.88% | 68.25% |
> | Q Proj | Sparsity (%) | 68.20% | 69.32% | 70.39% | 68.20% | 67.03% | 66.54% |
> |  | Max θ Rot. (°) | 0.7838/0.7826 | 0.6052/0.6059 | 0.4606/0.4593 | 0.7838/0.7826 | 0.5968/0.5974 | 0.7618/0.7628 |
> |  | Max σ Diff | 1.99E-04 | 2.23E-04 | 9.50E-05 | 1.99E-04 | 2.16E-04 | 1.30E-04 |
> | O Proj | Sparsity (%) | 56.17% | 57.49% | 59.56% | 56.17% | 54.84% | 54.81% |
> |  | Max θ Rot. (°) | 3.2303/3.2325 | 3.6667/3.6684 | 2.2130/2.2134 | 3.2303/3.2325 | 1.6049/1.6057 | 1.8305/1.8318 |
> |  | Max σ Diff | 1.00E-04 | 8.80E-05 | 1.12E-04 | 1.00E-04 | 9.70E-05 | 9.30E-05 |
> | MLP Up | Sparsity (%) | 62.98% | 64.35% | 66.10% | 62.98% | 61.51% | 61.51% |
> |  | Max θ Rot. (°) | 3.4001/3.3988 | 2.2517/2.2500 | 2.9038/2.9038 | 3.4001/3.3988 | 2.2094/2.2078 | 2.6206/2.6196 |
> |  | Max σ Diff | 1.23E-04 | 1.51E-04 | 1.56E-04 | 1.23E-04 | 2.14E-04 | 1.47E-04 |
> | MLP Gate | Sparsity (%) | 63.63% | 65.12% | 66.68% | 63.63% | 62.06% | 62.01% |
> |  | Max θ Rot. (°) | 2.6317/2.6305 | 1.8464/1.8454 | 1.2666/1.2649 | 2.6317/2.6305 | 3.6245/3.6235 | 2.0907/2.0893 |
> |  | Max σ Diff | 1.72E-04 | 1.55E-04 | 2.02E-04 | 1.72E-04 | 3.30E-04 | 1.27E-04 |

---

> ### Author Response · Authors · 2025-11-26
> **Response to Reviewer jERv part3**
>
> ## Q.4 Generalization Broadened (LLaMA)
> We have expanded our evaluation to include Llama.
>
> **Quantitative Mechanism:** We incorporated a Llama-3.2-3B into all new experiments (Precision Sweep in Table R1 and KL Sweep in Table R2). In all cases, it exhibits the same structural learning bias (off-principal steering and spectral preservation) as Qwen.
>
> **Visual Consensus:** Llama displays the same "stripe-like" update consensus across different algorithms, as visualized in the updated Figure 14.
>
>
> ------
> **Reference**
>
> [1] Nov 12, FP16 training and inference support, https://github.com/volcengine/verl/pull/4036
>
> [2] Liu, Zihang, et al. "LIFT the Veil for the Truth: Principal Weights Emerge after Rank Reduction for Reasoning-Focused Supervised Fine-Tuning." ICML 2025
>
> [3] He, Zhiwei, et al. Deepmath-103k: A large-scale, challenging, decontaminated, and verifiable mathematical dataset for advancing reasoning. arXiv preprint arXiv:2504.11456, 2025.
>
> [4] Hendrycks, , et al. "Measuring mathematical problem solving with the math dataset.""  NeurIPS, 2021.
>
> [5] Meng, Fanxu, Zhaohui Wang, and Muhan Zhang. "Pissa: Principal singular values and singular vectors adaptation of large language models." NeurIPS 2024

---

### Author Response · Authors · 2025-11-26
**General Response: Contribution Clarification & Summary of Revisions**

# General Response:

We thank all reviewers for their detailed feedback and constructive suggestions.

## Strengths & Consensus

We are encouraged that reviewers generally view our papar as *“fresh and thought-provoking”* (BtSB, jERv), *mechanistically clear* (jERv),  *strong empirical evidence and causal interventions* (BtSB, 6ikK), and offering *practical value for geometry-aligned learning algorithms and RL-specific PEFT* (jERv, 6ikK).

We build on this positive consensus by clarifying our core contribution and summarizing the new experiments added to fully address reviewer's concern.


## Reclaiming the Contribution: "The Path Not Taken -- The first study yields how parameters evolve in a surprising structural format during RLVR."

We feel that some concerns (notably from Reviewer QWqF) interpret our work as being primarily about **sparsity** or "**KL making updates small"** (Gate I only). We recognize that the original title ("Why RL Updates Look Sparse") may have contributed to this misunderstanding. Here, we would respectfully assert that **sparsity** or **small update** is not our main claim, as clearly stated in Sec 2.3.

Our actual contribution is the **structural learning bias** of RLVR — *where* updates go in parameter space, not just *how big* they are. To avoid the focus being the symptom (sparsity) ,not the true mechanism, we have updated the title in the revision to:

> **“The Path Not Taken: RLVR Provably Learns Off the Principals.”**

This better captures the main contributions:

**(i) Step towards a white-box understanding of RLVR.**
We provide the first **parameter-level characterization** of RLVR’s learning dynamics. We **characterize and formalize a structural learning bias via our Three-Gate Theory**: for a fixed pretrained model, RLVR consistently steers updates toward specific **off-principal, low-curvature regions**, rather than uniformly shrinking or randomly distributing updates.

**(ii) Guidance for geometry-aware, RLVR-native learning algorithms.**
We establish that RLVR operates in a **distinct optimization regime** from SFT (which tends to target principal weights). Consequently, directly adapting SFT-era PEFT heuristics can be flawed, since they are overly aligned to SFT dynamics and do *not* honor RL’s optimization geometry. Our case studies on advanced sparse fine-tuning and LoRA variants illustrate this mismatch and motivate **geometry-aware, RLVR-native designs**.


## New version uploaded with extra experiments

We have added new experiments in rebuttal and manuscript:
- **Precision disentanglement (bf16 / fp16 / fp32 sweep)  - Q.1 in the Responce to jERv**
  We add a strict precision sweep. The results strength our claim that while fp32 removes the **visual sparsity** shown in bf16, the **structural bias** (spectral preservation and principal avoidance) persists in full fp32, reinforcing that **precision is a lens, not the cause**.
- **Broader evaluation (Agents, RLHF, Embodied AI) – Sec. 4.1.**
  We extend our diagnostics beyond math/coding LLMs to multi-turn agents, RLHF models, and embodied/VLA policies. All exhibit the same **structural learning bias** (off-principal updates with preserved spectra).
- **Insight → geometry-aware algorithms – Sec. 5.**
  We polish the “Insight” section to make the practical takeaway clearer: how our observations guide the design of **geometry-aware, RLVR-native algorithms**. We also add a new experiment showing how our theory **predicts the failure of PiSSA** (a LoRA variant highlighted as a promising next step in the Thinking Machines' LoRA blog) in RL, despite its success in SFT.
- **LLaMA models included in new experiments.**
  LLaMA-3.2-3B is now exhaustively added in the precision sweep and an **update-consensus** analysis (Appendix G.1), showing the same structural patterns as Qwen/DeepSeek.
- **New figures/tables.**
  All new plots and tables referenced above are in the appendix (pages 24–29).

---

### Meta-Review · Area_Chair_er8Q · 2026-01-07

**Summary:**

This paper studies an empirical phenomenon in RLVR fine-tuning of LLMs: large downstream gains despite parameter updates that appear sparse. The authors argue the visible sparsity is a byproduct of a persistent optimization bias (“implicit compass”) and formalize it via a Three-Gate Theory: (I) an on-policy KL-style proximal constraint, (II) steering induced by pretrained geometry toward “off-principal/low-curvature” directions with spectral preservation, and (III) bf16 precision acting as a lens that hides micro-updates. They support the narrative with parameter-space diagnostics (spectral drift, principal-subspace rotation, mask overlap/consensus) and interventions that disrupt geometry.

Overall, the submission is interesting and timely, but after considering the rebuttal, I still recommend reject primarily due to lack of polish and unclear positioning/contribution relative to existing literature and practice-oriented fine-tuning baselines.

**Reviewer Concerns:**

**What the rebuttal addressed well**

Precision as “lens” vs “cause”: authors added bf16/fp16/fp32 sweeps and argue the visual sparsity shrinks with precision while their geometry metrics persist.

Broader evaluation: added evidence beyond math/code LLMs (agents, RLHF, robotics/VLA; also LLaMA inclusion).

Practical takeaway (partially): Section 5 was reportedly reworked to emphasize “geometry-aware RL-native” design, plus a case study suggesting a mismatch of certain SFT-era PEFT heuristics (e.g., PiSSA-style principal targeting) under RL.

Reproducibility: they mention adding probe code/scripts to replicate key diagnostics.

**What remains problematic (reasons for rejection)**

Non-polished presentation / unclear core contribution
Figure captions and annotations contain typos and grammatical errors, several claims are framed strongly (“first”, “provably”) but—based on the discussion text—do not clearly correspond to formal theorems or sharp, falsifiable statements. This leaves the contribution feeling under-specified.

Positioning vs. literature remains incomplete

Review requests to better situate the work in implicit bias/optimization geometry and prior RL sparsity/subnetwork findings appear only partially addressed.

Some rebuttal responses are dismissive (“factually incorrect premises”) rather than integrating the requested context into the paper. Even if the authors are right on specifics, the paper still needs a stronger, calmer literature discussion that helps readers map what is new.

Insufficient comparison to state-of-the-art fine-tuning practice

While the rebuttal adds selected case studies (e.g., LoRA variants, sparse masks), it still does not convincingly benchmark against current SOTA fine-tuning / PEFT choices used in RLVR/RLHF settings (beyond a narrow set of variants).

The “algorithm design guidance” is not backed by a comprehensive or compelling empirical story that demonstrates improved methods over strong baselines; it remains closer to an interpretation paper than a methods paper, but the submission’s framing suggests actionable guidance.

Key mechanistic link remains somewhat inferential.

The “principal weights = high curvature” link is defended largely via proxy arguments and prior work, but remains not fully validated in the submission (e.g., limited direct curvature/Fisher evidence, per reviewers’ original requests).

As a result, the mechanistic explanation still feels suggestive rather than decisive.

**Reviewer Scores:**

jERv: 8 → 8.
Their main requests (strict fp32 sweep; LLaMA replication; KL/clipping sensitivity) appear largely addressed.

BtSB: 4 → 4
They explicitly wanted clearer “tangible algorithmic guidance” and broader generalization.

6ikK: 4 → 4 or 6
Precision sweep + added reproducibility artifacts directly address core weaknesses.

QWqF: 2 → 2
This reviewer was negative; Even with clarifications, their score likely remains low, as some of their concerns were not addressed.

---

### Decision · Program_Chairs · 2026-01-26

Reject